

# Automatic generation of explanations in autonomous systems: enhancing human interaction in smart home environments

Oscar Peña-Cáceres[1], Antoni Mestre[2], Manoli Albert[2], Vicente Pelechano[2] and Miriam Gil[1]

[1] Departament d'Informàtica, Universitat de València, Valencia, Spain
[2] VRAIN Institute, Universitat Politècnica de València, Valencia, Spain

## ABSTRACT

In smart environments, autonomous systems often adapt their behavior to the context, and although such adaptations are generally beneficial, they may cause users to struggle to understand or trust them. To address this, we propose an explanation generation system that produces natural language descriptions (explanations) to clarify the adaptive behavior of smart home systems in runtime. These explanations are customized based on user characteristics and the contextual information derived from the user interactions with the system. Our approach leverages a prompt-based strategy using a fine-tuned large language model, guided by a modular template that integrates key data such as the type of explanation to be generated, user profile, runtime system information, interaction history, and the specific nature of the system adaptation. As a preliminary step, we also present a conceptual model that characterize explanations in the domain of autonomous systems by defining their core concepts. Finally, we evaluate the user experience of the generated explanations through an experiment involving 118 participants. Results show that generated explanations are perceived positive and with high level of acceptance.

## INTRODUCTION

The emergence of autonomous systems (ASs) has significantly impacted various aspects of daily life, including healthcare, smart home technology, vehicles, and more (*Maggino, 2014*). Although autonomous systems operate with a high level of automation, their effectiveness both now and in the foreseeable future is based on a collaborative framework where humans and machines work together, as full autonomy remains unattainable. This collaboration enables the optimal fusion of human expertise and machine capabilities to tackle complex challenges, ensuring reliable performance. For seamless human-system collaboration, individuals must perceive autonomous systems as both comprehensible and trustworthy (*Li, Zhang & Chen, 2020*). System explanations serve as vital tools to foster trust and enhance collaboration between humans and machines. In this work, an "explanation" refers to a natural language clarification provided by the system, either to

Corresponding author
Oscar Peña-Cáceres,
osjmarpe@alumni.uv.es

inform the user about an action the system has performed or to communicate what the system expects from the user. For example, the system may explain that it has activated irrigation due to a detected moisture deficit, or notify the user that it is awaiting confirmation to proceed with an automatic purchase. These explanations elucidate the actions and expectations of the system, empowering users to understand and engage with the system effectively.

Our vision is to enable the development of autonomous systems that can–at run-time–answer questions about their behavior, *e.g.*, why a certain action was taken, why the user has to collaborate with the system, how the user can collaborate, *etc.*

Achieving this vision requires explanations that are dynamic, adapting to changing circumstances and user contexts. Moreover, balancing the depth of information with simplicity is crucial, as much information on explanations can overwhelm users, while overly simplistic ones may fail to convey the system's intricacies adequately. Additionally, ensuring that explanations are timely and contextually relevant adds another layer of complexity to the development of autonomous systems that provide these types of explanations.

To address these challenges, autonomous systems can leverage artificial intelligence to infer the most appropriate characteristics of explanations based on the specific situation of the user and the system, and dynamically generate explanations accordingly. In this article, we first present a conceptual model to characterize explanations in the context of autonomous systems, and then we propose enhancing smart home systems by integrating a component designed to generate explanations that help users understand the adaptive behavior of the system. This component is developed using advanced techniques that fine-tune large language models (LLMs) with domain-specific data and prompts. By using this component, the system produces explanations that directly address and clarify its adaptive actions. These explanations, sensitive to contextual nuances, bridge the gap between the complexities of system operation and user understanding, thereby promoting transparency and trust. While the proposal is designed for a smart home system, the solution can be applied across a wide range of domains.

We validated the proposal by applying the short version of the user experience evaluation questionnaire (UEQ-S). This questionnaire was administered to a diverse group of users, including individuals with varying levels of technological experience and a balanced mix of genders and ages, to ensure a comprehensive evaluation of the explanations generated by our approach.

In this way, the main contribution of the article is a technique for generating explanation content based on LLMs for the smart home domain whose effectiveness has been validated through an experiment.

The rest of the article is structured as follows. 'Related Work' discusses recent advances and major contributions to the field of explanation generation. 'A Conceptual Model for Explanation Specification' introduces a conceptual model for defining explanations within this work. This section clarifies our vision of what explanations entail in the context of this article. 'Proposal Overview' introduces the proposal for explanation generation. In 'Smart Home LLM-Based Explanation Generator', we apply our proposal to build a generator for

explanations in the context of a smart home system. 'Generation of Explanations for the Smart Home' shows the result of applying our proposal in the case of the smart home system. The proposal is validated in 'Validation of the Proposal', and the user experience is evaluated in the "Evaluation of the User Experience". Some limitations of the current work are presented in 'Limitations'. Finally, 'Conclusions' presents conclusions and future work.

# RELATED WORK

Explanation generation in autonomous systems has garnered significant attention from researchers in various domains with the aim of enhancing user understanding, trust, and acceptance of these systems. In this section, we review recent advances and key contributions in the field of explanation generation. We begin by examining the domain of smart home environments focusing on approaches that base on LLMs. Then, we explore other domains that use LLMs and other techniques, evidencing traditional methods for explanation generation. We also identify previous work that addresses the integration of explanatory modules within system architectures.

## Explanation generation in the smart home domain

In smart home environments, where systems must often make sense of user activities and communicate their actions meaningfully, the generation of explanations remains a key challenge. In recent years, the rapid advancement of LLMs has led to the emergence of numerous approaches to generate explanations based on these models. LLMs have gained traction for their versatility in all domains, including tasks such as synthetic data generation and domain-specific fine-tuning, as highlighted in the survey by *Zhao et al. (2025)*. The adaptability of these models, such as GPT-4, allows them to tackle a wide range of tasks across domains, from summarization to interactive smart systems, offering a significant advantage in creating personalized and dynamic user experiences. Using vast amounts of data, LLMs can generate highly coherent and contextually relevant explanations, making them invaluable in areas such as smart home management.

Furthermore, *Yang et al. (2024)* investigate the broader application of LLMs in various domains, particularly focusing on synthetic data generation. They emphasize that fine-tuned models, adapted to specific domains, outperform general-purpose models for traditional tasks. While this reinforces the importance of domain-specific tuning (something we leverage in our own approach) their study does not address the challenge of providing real-time, context-aware explanations in an interactive setting.

A recent study by *Sadeghi et al. (2024)* emphasizes user-centered explanations, proposing a framework that integrates LLMs with context-aware techniques to generate explanations tailored to individual user needs. This approach is a step forward in creating personalized user interactions, using context to provide explanations that align with the user's immediate situation and preferences. However, their focus remains on static contexts rather than dynamically adapting explanations in real time as conditions change.

Similarly, *Duan, Li & Li (2024)* introduced ContextualHomeLLM, a sophisticated language model designed to enhance user interactions in smart homes by optimizing management tasks. This model offers personalized recommendations and real-time

responses, integrating environmental factors and historical data to accurately interpret user behavior and intentions. While ContextualHomeLLM demonstrates impressive advances in smart home management, the primary focus is on optimizing system actions and user recommendations rather than generating adaptive explanations that evolve based on continuous interaction and feedback from the user.

By the same token, *Rivkin et al. (2025)* propose a framework of autonomous LLM agents capable of executing and explaining smart home actions *via* natural reasoning. *Hulayyil, Li & Saxena (2025)* apply natural language justifications to home intrusion detection, improving user comprehension. *Das et al. (2023)* introduce an explainable activity recognition system leveraging SHapley Additive exPlanations (SHAP) and Local Interpretable Model-agnostic Explanations (LIME) to produce intelligible explanations. Unlike these works, our approach emphasizes behavioral regularity and tailors explanations to the context of the user, rather than focusing on isolated actions or single device contexts.

Other contributions also address smart home environments from a broader, system-level perspective. *Chen, Chen & Jin (2024)* employ LLMs to generate interpretable models of smart device behavior based on textual requirements, while *Sarhaddi et al. (2025)* provide a comprehensive survey on explainable artificial intelligence (AI) for Internet of Thing (IoT) systems, including smart homes, emphasizing adaptive and context-aware explanations. Unlike approaches that simply generate generic answers, our system adjusts its explanatory behavior based on historical patterns learned from data in the home environment. In this way, the explanations reflect the operational logic of the smart home by aligning with user expectations and actual system dynamics.

Another relevant contribution comes from *Civitarese et al. (2025)*, who explore the use of LLMs for developing a sensor-based recognition system for activities of daily living (ADLs) in smart homes. Their system, ADL-LLM, transforms raw sensor data into textual representations processed by an LLM to recognize ADLs efficiently. While this study underscores the potential of LLMs for improving functionality and user experience in smart homes, it focuses primarily on sensor-based recognition and does not address how explanations of system behavior are communicated to users in real time or how those explanations adapt dynamically to user feedback and changing contexts.

### Explanation generation in other domains

Beyond the smart home domain, explanation generation has been explored in various fields using interpretable models such as decision trees and rule-based systems, as well as *post-hoc* techniques like LIME and SHAP (as discussed in *Camilli, Mirandola & Scandurra (2022)*). These methods offer transparency, but struggle to adapt explanations in real time. MAPE-K loops, used in self-adaptive systems, allow users to query historical data and understand system behavior based on environmental changes. However, they lack the flexibility of dynamically generating context-specific explanations as user interactions evolve.

Recent work on human-in-the-loop systems adds another important layer to the discussion of dynamic and adaptive explanations. *Ullauri et al. (2022)* propose a

history-aware explanation approach for self-adaptive systems, where users can retrieve historical data about system behaviors and interact with the decision-making process in real time. This method extends the traditional MAPE-K loop by allowing users to query historical data and steer system decisions based on past interactions. This mechanism emphasizes transparency and trust, since users are actively involved in decision making, a crucial aspect that we also address in our approach. However, this work focuses more on giving users access to system histories than providing explanations tailored to the real-time interaction between the user and the system.

Similarly, *Magister et al. (2021)* explore concept-based explanations in graph neural networks (GNNs) using their GCExplainer tool. This tool allows users to understand complex model predictions by breaking them down into higher-level concepts, improving explainability and user trust. Although their focus is on GNNs, their approach shares similarities with ours in its goal of making explanations more comprehensible and user-friendly by leveraging abstract concepts. However, unlike our focus on dynamically adapting explanations in real-time to the evolving context of users, GCExplainer primarily deals with *post-hoc* explanations based on fixed concept representations.

## Architectural integration of explanation generation

While a significant portion of the literature focuses on the content and form of explanations generated by LLMs, less attention has been paid to how explanatory modules are architecturally integrated into adaptive systems. Understanding this integration is crucial for enabling runtime explanation generation that aligns with system behavior and contextual dynamics. *Houze et al. (2022)* proposes a modular architecture for self-explanatory smart homes. *Bencomo et al. (2010)* provides foundational insights on how reasoning modules, including those used for generating explanations, can be integrated and maintained at runtime. *Kim et al. (2024)*, meanwhile, address the integration of LLMs into intelligent robotic architectures, exploring their role in perception, reasoning, and control processes. Complementarily, *Shajalal et al. (2024)* emphasize the importance of incorporating explainability from the initial stages of user-centered design in smart home contexts. However, a gap persists around approaches that articulate explanatory generation with real-time system operational logic. In response to this challenge, the present work incorporates an automatic adaptive explanatory generation module as a functional component within the system architecture, enabling seamless operational integration tailored to the contextual dynamics of the smart environment.

Collectively, these studies highlight the increasing role of LLMs in improving functionality, personalization, and real-time responses within smart home environments. Each of these approaches demonstrates notable improvements in user interaction and system optimization by integrating various factors such as context-awareness, sensor data, and user history. However, they focus predominantly on optimizing system actions or providing static explanations, lacking mechanisms for real-time, adaptive explanation generation that respond dynamically to the specific needs, contexts, and interactions of users.

In contrast, our work distinguishes itself by focusing on the dynamic generation of personalized explanations at runtime. Unlike the approaches mentioned above, we address the challenge of continuous adaptation, where explanations evolve as the user's context, system behavior, and interaction history change. By using an advanced LLM fine-tuned with domain-specific data from the smart home environment, we enable the system to provide explanations that are not only contextually relevant but also adaptive in real time, adjusting to ongoing interactions between the system and the user.

## A CONCEPTUAL MODEL FOR EXPLANATION SPECIFICATION

This section introduces a conceptual model that identifies the main concepts that characterize explanations in autonomous systems and their relationships. The model focuses on explanations provided by the system to the user, offering customized clarifications about what the system is doing or what it expects from the user. These explanations are adapted to the user profile, the system context, and the user's historical behavior. This conceptual model underpins the proposal described in the next section. In particular, the explanations generated by our approach are constructed based on the concepts and relationships of the model. The initial version of the conceptual model was introduced in the work of *Mestre et al. (2022)*. Building upon that initial version, the present work refines the model to adapt it to autonomous systems. To this end, we have reviewed existing literature focusing on research in explainable systems in the context of smart environments. We reviewed well-established taxonomies of explanations, such as the work of *Chari et al. (2020)* which define an ontology of explanation primitives for user-centered AI, the work of *Lim, Dey & Avrahami (2009)* which defines a taxonomy for intelligibility in context-aware applications, and the work of *Nunes & Jannach (2017)* which summarizes several purposes for explanations in the context of AI systems.

Figure 1 shows the model, where orange boxes define essential elements of the explanations, blue boxes denote explanation features, and green boxes specify the system and user situation, as well as the user's historical behavior. The green elements collectively facilitate the identification of the precise situation in which the explanation will be given.

The orange boxes in Fig. 1 define conceptual elements that are basic for the specification of an explanation. The *explanation goal* represents the purpose or reason why a user needs an explanation (*Miller, Howe & Sonenberg, 2017*). Two main explanatory objectives are distinguished (*Gil et al., 2019*): "*feedback*," which explains actions performed by the system to inform or justify, and "*feedforward*," which involves user participation and explains actions to be taken by the user. For instance, in a virtual assistant system, a feedback explanation might notify the user about an automatic calendar update, while a feedforward explanation could entail soliciting the user's preference for scheduling a reminder. An explanation goal is targeted at a user (*human*) (*Chari et al., 2020*). The intention is to manage to get him/her to understand the system in order to collaborate with it. Achieving the explanation goal can be of great importance (or not) to the system. To specify this, the conceptual framework proposes the *criticality* concept associated with the explanation

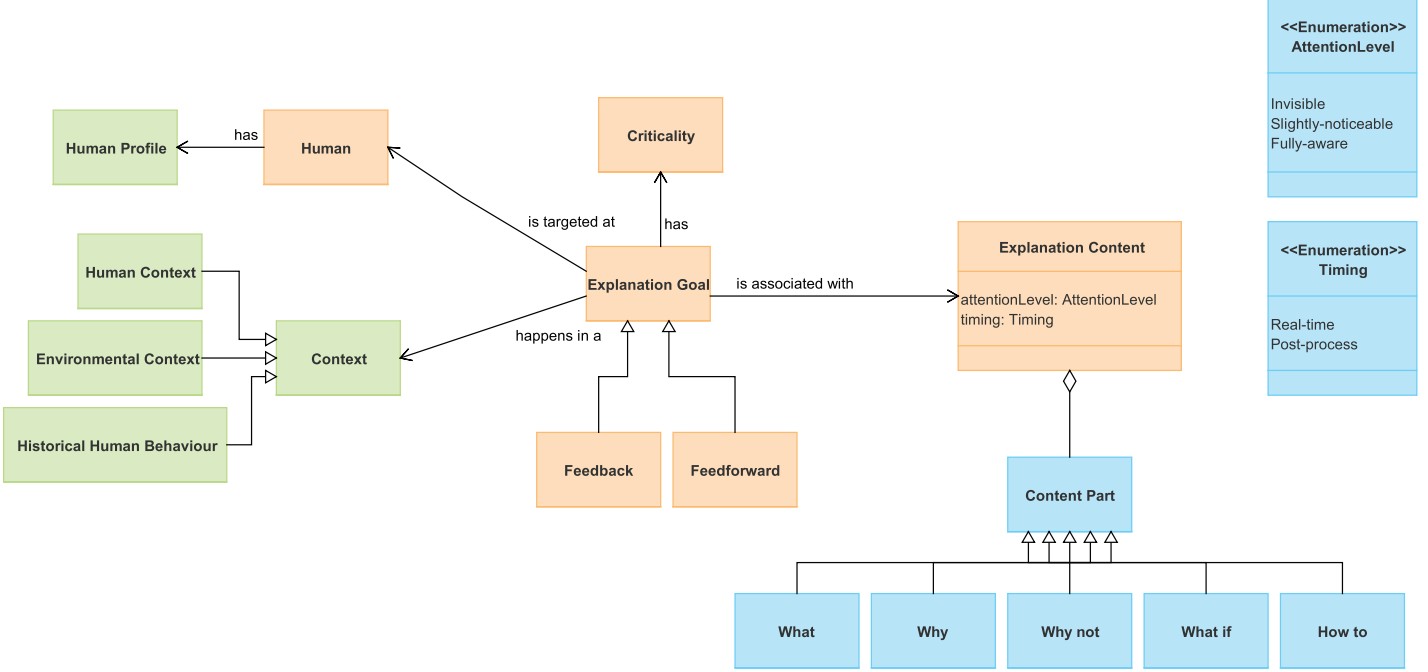

**Figure 1 Reference conceptual model to characterize explanations.**

goal. *Criticality* determines the extent to which the information provided in the explanation is essential for the user to achieve their goal (*Lim, Dey & Avrahami, 2009*). In addition, the *explanation content* is associated with an explanation goal. This concept encompasses the actual substance or information provided within the explanation to fulfill its goal. All these concepts constitute the foundation for the explanations.

The blue concepts in Fig. 1 characterize the features of the content of the explanations. According to the work of *Lim, Dey & Avrahami (2009)*, the explanation content can have different *content parts* depending on the question they answer:

- "What": what has the system done?
- "Why": why did the system do X?
- "Why not": why didn't the system do Y?
- "What if": what would the system do if W happened?
- "How to": how can I get the system to do Z, given the current context?

It is important to note that feedback explanations typically focus on retrospective clarification—they address "what" the system did and "why" it did it, based on past or ongoing actions. In contrast, feedforward explanations are prospective in nature—they guide the user toward future actions, often emphasizing "how to" proceed or "what if" alternatives. However, feedforward explanations still require some degree of feedback content, as users need to understand the current system state or the triggering condition in order to contextualize the suggested next step. For example, advising a user to reset a device requires first informing them of the malfunction that occurred. Therefore, while the

core function of feedback is justification and the core function of feedforward is instruction, effective feedforward explanations integrate feedback components to remain coherent and actionable. This interdependence reinforces the need for well-structured, hybrid explanatory messages in adaptive systems.

Furthermore, the conceptual model addresses two additional features related to when and how to present the explanation. It takes into account the *timing* attribute (*Glomsrud et al., 2019*) to determine whether the explanation occurs in real-time or during post-processing and considers the *level of attention* attribute to categorize the extent of attention required from users. Previous works (*Gil, Giner & Pelechano, 2012*; *Horvitz et al., 2003*) have investigated how attention management affects the ability of a user to cooperate with the system, and the need to control the level of attention required by an explanation. We have defined three levels: invisible, slightly-noticeable or fully-aware.

Finally, the green squares in Fig. 1 identify concepts to characterize the user context, as well as to provide the system context within an explanation (*Chari et al., 2020*). The aspects we consider are the user's profile, the human and environmental context, and historical human behavior. The *human profile* encompasses significant aspects to define the user such as "age", "technological capability", and "experience." The context of the *user* indicates any information relevant to characterize the situation of the user, such as location, activity, and device used. *Environmental context* encompasses external factors of the system's environment such as "date", "time", and "environmental conditions." Lastly, *historical human behavior* records previous human-system interactions, indicating errors and relevant user behaviors.

We apply the conceptual model to characterize an explanation within the smart home domain. This domain illustrates the types of explanations discussed in the article, demonstrating how these explanations are decomposed according to the proposed conceptual model.

Consider the following adaptive behaviour. The smart home has an irrigation service that has identified a malfunction in one of the sprinklers. As a precaution, the irrigation system will remain inactive until the user manually adjusts the sprinkler. Therefore, the system requires human intervention for the manual adjustment. This adaptive behavior may require an explanation. This would be an example of a feedforward explanation since the system explains actions to be taken by the user.

Suppose that the scenario in the smart home is the following: James, a 23-year-old man with advanced technological expertise is finishing a peaceful breakfast, enjoying a leisurely holiday morning. He is in a relaxed and receptive state of mind. In this situation, Table 1 shows an instance of the conceptual model for the feedforward explanation for the manual adjustment of the sprinkler. Based on this contextual information, in this article we address the generation of the explanation content. In this example, a suitable explanation content manually built ad-hoc for this scenario may be: "*Please make a manual adjustment to the sprinkler system's 'main' to restore it to proper operation, as it has exceeded 50 pounds per square inch (PSI), which is beyond the water pressure limit. Until this adjustment is made, the watering system will not restart.*". In next sections, we define how this contextual

**Table 1 Instance of the conceptual model for the introduced scenario.**

| | |
|---|---|
| Explanation goal | Wait for manual adjustment of sprinkler |
| **Type of explanation** | Feedforward |
| **Criticality** | Medium |
| **Human** | James |
| **Explanation content:** | |
| Timing | Real-time |
| Attention level | Low |
| Content part | What, why |
| **Human profile** | Age = 23 technological expertise = advanced |
| **Context:** | |
| Human context | Activity = breakfast; state = relaxed |
| Environmental context | Day = saturday; time = 9:15 a.m. |
| Historical human behavior | Interaction history = 0 |

information is used as an input to automatically generate the explanation content at runtime.

This framework establishes the basis for understanding explanations within autonomous systems. Building on this foundation, the article addresses the challenge of generating automated content that is both comprehensible and engaging for users. The primary objective is to produce explanations, grounded in this conceptualization, that enhance users' understanding of the system's adaptive behavior.

## PROPOSAL OVERVIEW

The proposal presented in this article aims to automatically generate explanations to help users understand when a system autonomously adapts its actions in response to changes. Such adaptation can often lead to user confusion, resulting in misunderstandings about the system's behavior, potentially causing incorrect responses and decreased user cooperation. Therefore, providing users with explanations in these circumstances is essential. Our solution generates natural language descriptions that are tailored to the user and their specific situation, taking into account their interaction history with the system.

The overall explanation process involves three steps: (1) detecting the need for an explanation, (2) generating a context-aware explanation, and (3) delivering it to the user through an appropriate interface. Steps 1 and 3 are addressed in previous work (*Mestre et al., 2022*) and ongoing research efforts, respectively. Although they are not the main contribution of this article, they are described to provide the necessary context. The contribution of this article focuses exclusively on step 2: generating a context-aware explanation based on user characteristics and situational context. In Fig. 2, step 2 is delimited with orange dashed lines to visually distinguish the scope of our contribution.

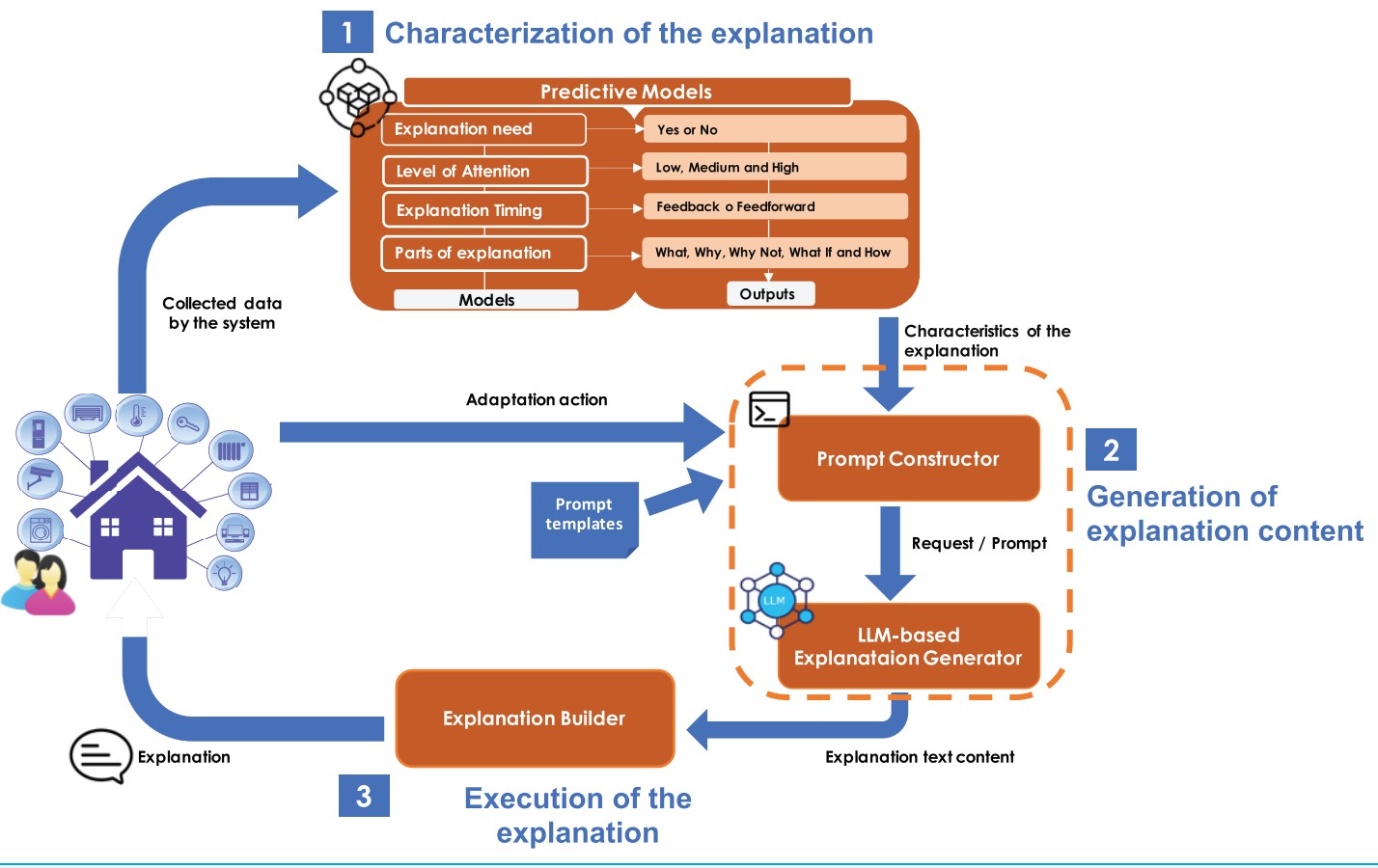

**Figure 2  Key components of the proposal.**

Figure 2 illustrates the architecture of the whole solution, showing all the components used for automatic explanation generation. The solution leverages a sensor-monitored system to collect data. Then, the solution operates as follows:

1. **Characterization of the explanation:** when the system performs an adaptation, it queries a predictive model to determine whether an explanation is necessary and, if so, what characteristics the explanation should have. This predictive model is designed to assess the need for an explanation based on information from the system, the user, and the history of interactions between the user and the system. If an explanation is required, the predictive model further infers the specific characteristics the explanation should exhibit. These characteristics define the necessary values for properties such as the level of attention required, the appropriate timing to provide the explanation, and the content elements that the explanation should include (these properties stem from the conceptual model—see Fig. 1). This predictive model has been built using supervised machine learning. Hosted remotely and accessible *via* application programming interface (API), the model receives contextual input and returns predictions in JavaScript Object Notation (JSON) format (see step 1 of Fig. 2). This step has been implemented in previous work (*Mestre et al., 2022*).

2. **Generation of the explanation content:** when the predictive model identifies the need for an explanation due to a system adaptation action, the system generates a request for explanatory content (a natural language description). This process involves:

- automatically constructing a prompt using a predefined template. The template is filled with data from the predictive model's output (explanation characteristics) and the system's log (which includes the executed adaptive action and the triggering event). This construction is managed by the component known as the *Prompt Constructor*, as shown in step 2 of Fig. 2.
- Processing the resulting prompt by the *LLM-based Explanation Generator* to produce the required explanatory content.

For the construction of this *LLM-based Explanation Generator*, we outline two key steps:

- Creating a domain-specific dataset: to facilitate the fine-tuning of the LLM, a domain-specific dataset must be created. This dataset should be tailored to the particular domain in question, ensuring that it contains relevant and accurate records that the model can use for training. In the following section, we detail the creation of such a dataset for the smart home domain.
- Fine-tuning the LLM: once the domain-specific dataset is prepared, the LLM undergoes a fine-tuning process. This involves re-training the model using the curated dataset to adjust its parameters and improve its performance in generating contextually appropriate explanations. The fine-tuning process ensures that the LLM is well-adapted to the specific needs of the domain, enabling it to produce more accurate and relevant explanations. In the following section, we illustrate this fine-tuning process with an example from the smart home domain.

The resulting *LLM-based Explanation Generator* is capable of producing explanations that align with the system's actions and the specific explanation characteristics determined by the predictive model of step 1. This enhances the overall quality and relevance of the generated explanations, making them more useful and understandable for users across various domains.

3. **Execution of the explanation:** once the explanation content is generated, the *Explanation Builder* selects the appropriate interaction mechanism for delivering the explanation, and the explanation is provided to the user (step 3 of Fig. 2). For this purpose, we propose using AdaptIO (*Gil & Pelechano, 2017*), a software infrastructure designed to adapt notification interactions based on the user's context. AdaptIO monitors the context and adjusts the interaction mechanisms of notifications (in our case, explanations) according to the level of attention. This step falls outside the scope of the present work and is proposed as future work.

It is important to note that our solution works for autonomous systems that use adaptation rules to define adaptive behavior. Specifically, an adaptation rule defines

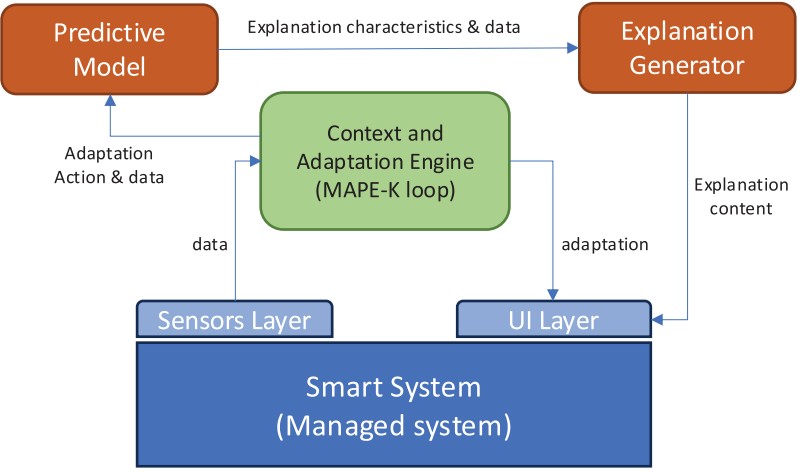

**Figure 3 Architecture of the solution.**

the actions to be taken in response to an event if a condition is met. An adaptation rule is defined as follows: *Event [Condition] Action*. For example, in a smart home system, if a rise in temperature is detected, leading to an increase of more than five degrees, the air conditioning would be activated. The adaptation rule, which we call "*SignificantTemperatureIncrease,*" would be defined as follows:

*TemperatureIncrease [Increase >5] ActivateAC.*

## Integration within a smart environment

The proposed explanation generation system is designed to operate as an integral part of a broader smart environment architecture. Figure 3 illustrates the global architecture in which the explanation module is embedded.

This ecosystem includes four main components:

1. Sensor layer: this layer collects real-time contextual data about the environment and the user. It includes motion sensors, ambient sensors (temperature, light, noise), and user-device interaction logs (*e.g.*, smart TVs, thermostats, lighting systems).

2. Context and adaptation engine: this module is responsible for analyzing sensor data to detect user activity patterns, emotional states (when possible), and triggering events that may require system adaptation. It also logs user preferences and system responses to maintain a history of interactions. This module can follow the architecture of MAPE-K loops for self-adaptive systems (*Rutten, Marchand & Simon, 2017*).

3. Explanation generator (our proposal): once a system adaptation is triggered, the predictive model determines whether an explanation is needed and what type of explanation is most appropriate. Our module then generates the explanation using the constructed prompt, guided by several input variables (explained above).

**Table 2 Key input variables used to generate context-sensitive explanations, including example values and data collection methods.**

| Category | Example value | Collection method | Description |
|---|---|---|---|
| User profile | Age: 55 | Provided during system setup | Demographic data and familiarity level with smart devices. |
| | Experience: Low | | |
| Device and appliance states | Refrigerator: ON | Smart home sensors and device status logs | Real-time status of connected appliances. |
| | Air conditioner: OFF | | |
| Environmental conditions | Temperature: 35 °C | Internal environmental sensors and weather services | Information about both indoor and outdoor conditions. |
| | Rain: 3 mm | | |
| Home activity indicators | Users at home: 1 | Presence sensors, pressure sensors, and door/window sensors | Indicators of movement, occupancy, and user activity in various home zones. |
| | Main-door: Closed | | |
| Security and automation systems | Camera: ON | Security sensors and automation logs | Data on surveillance, access, and automated home systems. |
| | Intruder detection: 0 | | |
| Historical user behavior | Interaction history: 0 | System logs | Number of times the user has not participated correctly in the service. |
| Attention level | Attention level: low | Output of the predictive model | Required attention level for the explanation. |
| Timing | Timing: real-time | Output of the predictive model | Appropriate timing to provide the explanation. |
| Content parts | Content: what, why | Output of the predictive model | Content parts that should contain the explanation. |
| Criticality | Criticality: medium | Output of the predictive model | Extent to which the information provided in the explanation is essential for the user. |

4. User interface layer: the generated explanation is delivered through a multimodal interface (*e.g.*, screen, voice assistant, mobile notification), adapted to the user's preferences and current activity.

To produce meaningful and context-sensitive explanations, the explanation generator relies on a set of key input variables that capture the user's profile, the real-time state of the smart environment, and the explanation characterization from the predictive model. These variables are derived from sensors, automation systems, and user-provided configuration data. Table 2 summarizes the main categories of input variables used by the system, along with representative values and the methods through which they are collected.

These variables are collected and processed by the context engine and passed to the predictive model when an adaptation is triggered by the adaptation engine. The predictive model determines the need of the explanation and the explanation characteristics (attention, timing, and parts) and pass this data to the explanation generator through the prompt constructor, which formats the information into structured inputs for the LLM to generate natural language explanations. By operationalizing the explanation process within a complete system architecture and grounding the inputs in real-time contextual data, the proposed approach becomes feasible for real-world deployment. This modular and interoperable design ensures the explanation system can be integrated into existing smart environments with minimal additional infrastructure.

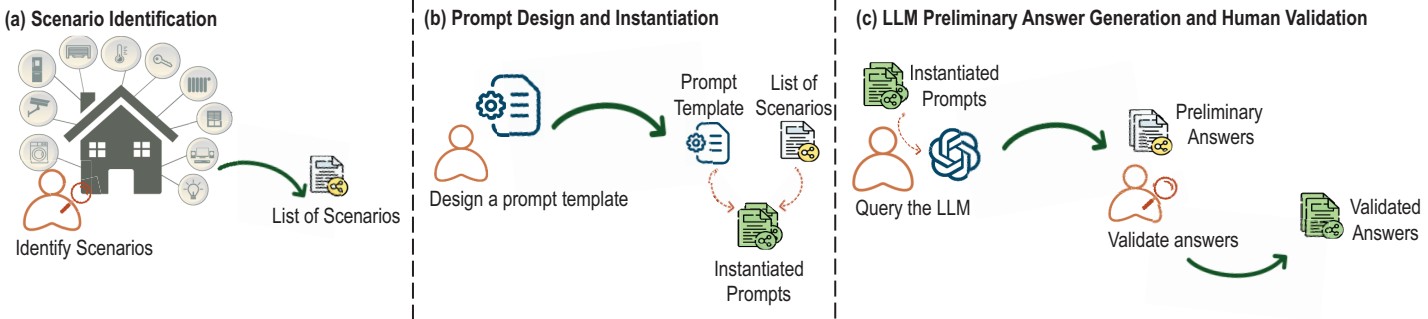

**Figure 4 Key steps of the dataset creation process.**

# SMART HOME LLM-BASED EXPLANATION GENERATOR

This section details the implementation of the *LLM-based Explanation Generator* for a smart home system. This component generates natural language descriptions tailored to specific scenarios in a smart home environment, enabling users to understand the adaptive actions taken by the system. The generator is one of the components that enables step 2 in Fig. 2, which illustrates the key elements of our approach. Its development, as mentioned in the Proposal Overview section, involves two main steps: (1) creating a domain-specific dataset for smart homes, and (2) fine-tuning a LLM to produce contextually appropriate explanations. These steps ensure that explanations are accurate, personalized, and responsive to the real-time context of the user and the system. Next subsections introduce the implementation of these two steps for the smart home system.

## Creation of a domain-specific dataset for smart homes

The development of the explanation generator begins with creating a dataset based on typical smart home scenarios. This dataset is used to train the LLM to generate relevant and precise explanations across various contexts. It consists of pairs of scenario-based prompts and their corresponding explanations. We have applied a structured methodology to create the dataset, as illustrated in Fig. 4 (*Zhuang et al., 2024*).

First, as depicted in part (a) of Fig. 4, we conducted a thorough identification of smart home scenarios where the system's adaptive behavior would need user explanations. These scenarios encompass a wide range of smart home functionalities, such as managing lighting systems, controlling climate settings, handling security protocols, and operating appliances. The focus of this identification process was to pinpoint situations where users might require clarification regarding the system's actions or guidance on what steps to take next.

Each scenario is designed to address one of two explanation types: feedback (explaining why the system performed a particular action) and feedforward (explaining what the user should do next). For example, in a feedback scenario, the system might explain why it turned off the heating due to an open window, whereas in a feedforward scenario, the system might instruct the user to close the window in order to resume heating.

To ensure the dataset's diversity and comprehensiveness, we included scenarios that span routine actions (*e.g.*, automatically adjusting lighting based on time of day) as well as exceptional cases (*e.g.*, responding to device malfunctions or security breaches). This variety ensures that the system can generate explanations that are not only accurate but also contextually tailored to the user's specific situation, whether it's a standard operational adjustment or an urgent issue requiring immediate attention.

After this identification process, we compiled a total of 228 distinct scenarios. This scenario set forms the foundation of the dataset used for training the explanation generation model, ensuring that it can handle diverse and real-world user interactions effectively.

Next (part (b) of Fig. 4), we designed a structured prompt to query the LLM, specifically GPT-3.5-turbo-1106, and generate detailed explanations. The prompt template ensures that each explanation is contextually appropriate and detailed. The structure of the prompt template is the following:

```
Produce an explanation for {Action} triggered by {Event} [Condition:
{Condition}] of type {Type} with an attention level {Level},
containing {Explanation Content} (What, Why, etc.), using {Log Data}
and {User Profile}.
```

Each element in the prompt serves a specific purpose:

- **Action:** specifies the system's action (*e.g.*, "turned off the irrigation system").
- **Event:** describes the event that triggered the system action (*e.g.*, "high water pressure").
- **Condition:** (optional) defines any specific conditions under which the event occurred.
- **Type:** indicates whether the explanation is feedback (explaining past actions) or feedforward (explaining future steps). It represents the primary focus of the explanation, not an exclusive category since feedforward explanations also include a minimal feedback component.
- **Level:** refers to the required attention level (low or high) based on the user's current state or needs.
- **Explanation content:** specifies the content to be included in the explanation, such as "what happened" or "why it happened."
- **Log data:** includes relevant system logs and data associated with the action or event. System logs include information about the device and appliance states, environmental conditions, home activity indicators and security and automation systems as defined in Table 2.
- **User profile:** incorporates user-specific data, such as preferences, technological experience, or interaction history.

The prompt template is instantiated by the identified scenarios. To do this, the specific details of the identified scenarios, including the system actions, triggering events, and relevant conditions are incorporated to the prompt template. This process ensures that each prompt is accurately tailored to reflect the unique context of each scenario. This set of

instantiated prompts serves a dual role: initially, it enables the generation of preliminary responses, and subsequently, it constitutes the input, as structured questions, for the dataset.

Once the prompts were instantiated with scenario-specific data, they were processed by the LLM (as shown in part (c) of Fig. 4). At this stage, the LLM generated preliminary explanations for each scenario. Traditionally, an expert would manually generate these outputs, carefully crafting responses to ensure clarity and relevance. However, in this LLM-assisted methodology, the LLM provides these preliminary explanations, which are then subjected to human validation. During this review, experts rigorously assess each response for accuracy, clarity, and contextual relevance.

This iterative refinement process is crucial for ensuring the quality of the generated explanations. Explanations that do not meet the required standards are adjusted by domain experts. This cycle of generation and review continues until the explanations are fully validated. Through this iterative process, we generated a dataset of 228 instantiated-scenario prompt and response pairs.

## Fine-tuning using ChatGPT 3.5

The fine-tuning process is a critical step in adapting a pre-trained language model to perform efficiently in a specific domain or task. In this work, we fine-tuned ChatGPT 3.5 to generate contextually appropriate explanations for smart home environments. The choice of ChatGPT 3.5 was motivated by its advanced capabilities in natural language understanding and generation, which is essential for producing high-quality, coherent explanations that address users' specific needs and queries. Additionally, the model's user-friendly API facilitated seamless integration and experimentation, further enhancing the efficiency of the fine-tuning process. The selection of GPT-3.5-turbo as the foundational model for our approach was guided by a balance between performance, accessibility, and fine-tuning capabilities at the time of implementation. While the landscape of large language models is evolving rapidly, with promising alternatives such as Mistral, LLaMA 2/3, Claude 3, or Gemini, many of these models presented constraints in one or more critical aspects of our study. First, GPT-3.5-turbo offered a mature and well-documented fine-tuning API, enabling stable integration into our development pipeline and the ability to train on domain-specific data. In contrast, open-source models such as Mistral or LLaMA often require substantial local infrastructure for fine-tuning, including graphics processing unit (GPU) clusters and parameter-efficient training strategies, which can limit reproducibility and portability in constrained environments like smart homes. Claude and Gemini, while competitive in reasoning, lacked at that time either fine-tuning availability or offered limited integration capabilities through public APIs. Furthermore, GPT-3.5 ensured strong performance in natural language generation tasks and benefited from widespread community adoption and tooling support, reducing integration overhead.

The fine-tuning procedure was carried out using the domain-specific dataset created for the smart home domain. This dataset consists of question-answer pairs, where each question represents a prompt instantiated with a specific scenario where each question
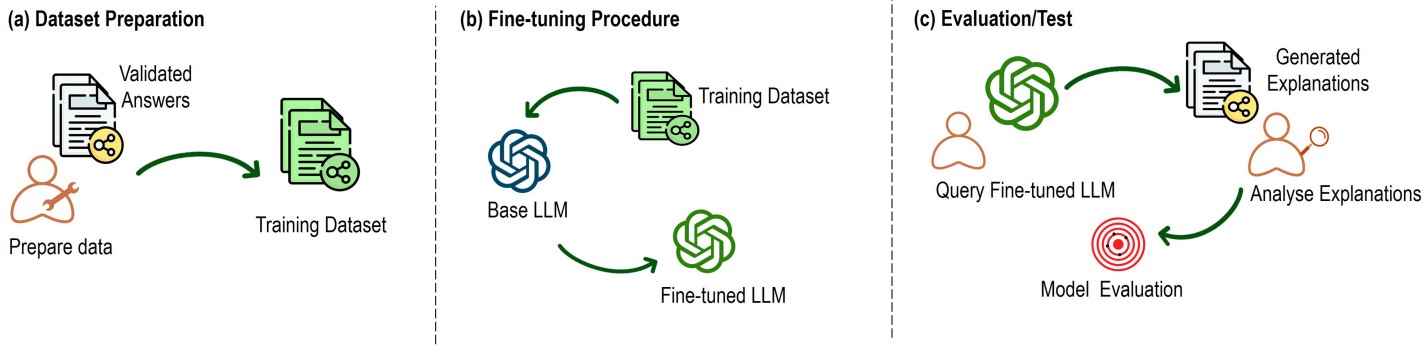

**Figure 5 Key steps of the fine-tuning process.**

represents a prompt instantiated with a specific scenario, and the corresponding answers were validated by human experts to ensure both accuracy and relevance. The primary goal of fine-tuning was to optimize the model's ability to generate accurate, context-sensitive explanations within the smart home environment.

The fine-tuning process followed a systematic methodology, illustrated in Fig. 5, which is based on three main stages: dataset preparation, fine-tuning procedure, and evaluation.

The first stage involved preparing the dataset (part (a) of Fig. 5), a foundational step to ensure that the model can accurately learn and respond to domain-specific interactions in the smart home context. In the previous section, we created a dataset of 228 scenarios. However, additional data preparation steps were essential to ensure the dataset was optimally suited for fine-tuning.

Initially, we conducted a thorough data cleaning process to ensure consistency in formatting across entries, checking that each scenario adhered to a standardized structure without any unintended variations. Given that the dataset was designed to minimize ambiguity from the outset, this cleaning focused on reinforcing uniformity across fields like action, triggering event, condition, and explanation type, without altering the carefully designed scenario content.

Then, to maximize the model's learning efficiency, the dataset was split into training and test sets with an 85:15 ratio. This approach allowed the model to train on 194 scenarios, encompassing a comprehensive range of smart home explanations, while setting aside 33 scenarios for final testing. This split supported both robust learning and reliable evaluation, ensuring that the model could generalize effectively across unseen scenarios.

The second stage of the process (part (b) of Fig. 5) involved fine-tuning the model with the prepared dataset. The base model, GPT-3.5-turbo-1106, was initialized with its pre-trained knowledge and then fine-tuned using the smart home dataset. This fine-tuning process involved adjusting the model's weights across multiple training iterations to enable it to learn patterns specific to the smart home context. These iterations were critical for optimizing the model's performance, ensuring that it could generate accurate, context-sensitive, and coherent explanations when responding to smart home-related queries.

For this evaluation, we used the test set of 33 scenarios that were set aside during data preparation, allowing us to assess the model's generalization ability on unseen examples.

The fine-tuning was conducted with a batch size of four and across seven epochs. The number of epochs was selected to provide an optimal balance between learning from the dataset without overfitting. During each epoch, the model was exposed to the full dataset, iteratively refining its parameters based on the domain-specific data.

Finally, the model was evaluated to measure the effectiveness of the fine-tuning process (part (c) of Fig. 5). We employed the explanation consistency metric (*Zhuang et al., 2024*), a key metric used to assess the reliability and coherence of the model's responses. This metric quantifies the consistency of model-generated explanations across related questions, crucial for ensuring coherent and reliable responses in practical applications.

The explanation consistency metric measures the percentage of responses that maintain consistency with a user's expectations across a set of related queries. Specifically, it quantifies the proportion of answers that align with the initial explanation provided for a given question, considering variations in follow-up questions. A score of 1 indicates perfect consistency, where the model's responses remain unchanged across all related examples.

For this evaluation, we used the test set of 33 scenarios that were set aside during data preparation, allowing us to assess the model's generalization ability on unseen examples. The fine-tuned ChatGPT 3.5 model achieved an average consistency score of 0.92. This result demonstrates that the model is highly proficient in maintaining coherence and generating contextually appropriate explanations across diverse smart home scenarios. The high consistency score highlights the model's capability to deliver reliable and meaningful explanations, which are essential for enhancing user interaction and satisfaction in smart home environments.

Fine-tuning using a domain-specific dataset has proven to be an effective approach for improving the model's ability to generate natural language explanations. The systematic methodology of dataset preparation, iterative training, and rigorous evaluation ensures that the fine-tuned model can provide contextually relevant, coherent, and accurate explanations that align with users' needs and the operational dynamics of a smart home. The successful application of this fine-tuning process marks a significant step toward developing more intuitive and responsive systems in the realm of smart home automation.

## GENERATION OF EXPLANATIONS FOR THE SMART HOME

In this section, we demonstrate the functionality of a smart home system developed by implementing the solution proposed in the previous sections. As a result, the smart home system is capable of automatically generating explanations for its adaptive behavior. That is, the solution automatically generates explanations in response to the adaptation actions performed by the smart home.

To demonstrate the solution in action, we present three smart home scenarios with adaptive behavior. In each scenario, the system queries the predictive model to determine whether an explanation is needed and what characteristics it should have. If an explanation is required, the system generates its content. As shown in step 2 of Fig. 2, the generation of the explanation content involves:

**Table 3 Generated explanations for five adaptive behaviour scenarios.**

| Scenario | Explanation setup | Generated explanation |
|---|---|---|
| A | **Explanation characteristics:**<br>• High level of attention<br>• Providing now the explanation<br>• Including the "What", "Why", and "How to" parts.<br>**Prompt:**<br>Produce an explanation for *wait-for-manual-reset* triggered by *heating-thermostat* [under the fulfillment of this condition *Temperature ≤14 $C*] of type *Feedforward* with an attention level *High* containing the *What, Why*, and the *How to* with the following information: LOG* + User Profile*. | *The system turned off the heating because it detected a problem with the thermostat. To turn it back on, you need to manually reset the thermostat.* |
| B | **Explanation characteristics:**<br>• Low level of attention<br>• Providing later the explanation<br>• Including the "What" part.<br>**Prompt:**<br>Produce an explanation for *false-alarm-trigger* triggered by *motion-detection-error* [under the fulfillment of this condition *No-visible-treat && intruders=0*] of type *Feedback* with an attention level *Low* containing the *What* with the following information: LOG* + User Profile*. | *The alarm was triggered by movement from the guest bedroom window sensor. No suspicious activity was detected.* |
| C | **Explanation characteristics:**<br>• Low level of attention.<br>• Providing later the explanation.<br>• Including the "What" and "How to" parts.<br>**Prompt:**<br>Produce an explanation for *wait-for-manual-adjustment* triggered by *High-pression-sprinklers* [under the fulfillment of this condition *Pressure-sprinkler[all] >50 PSI*] of type *Feedforward* with a level of attention *Low* containing *What* and the *How to* with the following information: LOG* + User Profile*. | *The main sprinkler is off due to pressure above 50 PSI. Manually adjust the pressure valve until it drops below 50 PSI and restart the irrigation system to resume operation.* |
| D | **Explanation characteristics:**<br>• Medium level of attention.<br>• Providing now the explanation.<br>• Including the "What" and "How to" parts.<br>**Prompt:**<br>Produce an explanation for *irrigation-postponed* triggered by *weather-forecast* [under the fulfillment of this condition *Rain-probability >85%*] of type *Feedforward* with a level of attention *Medium* containing *What* and the *How to* with the following information: LOG* + User Profile*. | *Rain is expected in the next few hours. You can skip the scheduled watering by updating the settings in the system.* |
| E | **Explanation characteristics:**<br>• Low level of attention.<br>• Providing now the explanation.<br>• Including the "What" and "What if" parts.<br>**Prompt:**<br>Produce an explanation for *ventilation-recommendation* triggered by *low-humidity* [under the fulfillment of this condition *Heater-on >3 h && Humidity <30%*] of type *Feedforward* with a level of attention *Low* containing *What* and the *What if* with the following information: LOG* + User Profile*. | *The heater has been running for a while. If the heater keeps running, humidity may drop further, and you would have to ventilate the room to maintain comfort.* |

- constructing a prompt by instantiating the template proposed in the previous section based on the specific situation, and
- querying the LLM-based explanation generator, developed in the previous section, using the constructed prompt.

The scenarios are the following:

- **Scenario A:** the smart home system has detected a malfunction in the heating thermostat and has shut down, awaiting manual adjustment. It's 7:30 p.m. in the middle of winter, and the house is occupied by Maria, a 65-year-old woman with limited technical knowledge. She has just arrived home after a long day at work.
- **Scenario B:** an alarm was triggered due to movement near the bedroom window. However, the smart home system did not detect any other suspicious activity. At 3:45 p.m., the only person at home is a 41-year-old man with intermediate technological knowledge. He has just finished lunch and is now cleaning the pool while listening to relaxing music.
- **Scenario C:** the smart home system has detected a malfunction in one of the sprinklers, as the pressure has exceeded maximum value. Consequently, it is shutting down the irrigation system until the issue can be resolved. The user interacting with the system is a 23-year-old man with advanced technological expertise. It is 9:15 a.m., and he has just finished a peaceful breakfast, enjoying a leisurely holiday morning. He is in a relaxed and receptive state of mind.
- **Scenario D:** it is 7:00 a.m., and Carolina, a 45-year-old woman with intermediate technology skills, is in the kitchen preparing her coffee. As she does every morning after breakfast, she usually confirms the watering of the garden, which she does between 7:20 and 7:40 a.m. However, the system has detected that it will rain for the next few hours.
- **Scenario E:** it's 5:30 p.m. and Andres, a 68-year-old man with basic technological skills, has just arrived home from his routine at the gym. Fifteen minutes earlier, his wife Maria went out and left the living room heater, which has been running since 2:00 p.m., on. The system detects that the ambient humidity has decreased and recommends ventilating the room.

For each scenario, Table 3 details: the characteristics of the explanation inferred by the predictive model, the constructed prompt, and the content of the explanation generated by our LLM-based explanation generator. The generated explanations demonstrate how our system produces textual content to explain the system's adaptive behavior. This content is tailored to the user's profile, the system's context, and the history of interactions between the user and the system.

## VALIDATION OF THE PROPOSAL

In this section, we present a comparative experiment evaluating our proposed explanation generation method against an alternative approach. This experiment aims to determine whether the combination of structured prompting and domain-specific fine-tuning leads

**Table 4  Configuration of models and prompting strategies used in the comparative evaluation.**

| Prompt type | Model | Description |
| --- | --- | --- |
| Simple prompt | GPT-3.5 (Generic LLM) | Using GPT-3.5-turbo with a minimal prompt that included only the description of the adaptation action. |
| Simple prompt | Claude 3 Haiku (Generic LLM) | Using Claude 3 Haiku with the same minimalistic prompting strategy. |
| Structured prompt | GPT-3.5 (Generic LLM) | Using GPT-3.5-turbo with the structured prompt described in Section "Proposal Overview", incorporating user profile, system context, and explanation content specifications. |
| Structured prompt | Fine-tuned LLM (our approach) | Using the domain-specific fine-tuned model developed in Section "Smart Home LLM-based Explanation Generator", combined with the structured prompt. |

**Table 5  Comparative evaluation results across different generation configurations.**

| Condition | Clarity | Useful | Adapted to the situation |
| --- | --- | --- | --- |
| Simple prompt—Generic LLM (GPT-3.5) | 3.1 | 2.9 | 2.7 |
| Simple prompt—Generic LLM (Claude 3 Haiku) | 3.2 | 3.0 | 2.8 |
| Structured prompt—Generic LLM (GPT-3.5) | 3.8 | 3.7 | 3.5 |
| Structured prompt—Fine-tuned LLM (our approach) | 4.4 | 4.5 | 4.6 |

to objectively better explanations compared to more generic or minimally guided generation strategies. To this end, we conducted a comparative study using different configurations of language models and prompt types, as detailed in Table 4. Unlike the subsequent section, which evaluates user perceptions in realistic scenarios, this experiment isolates the explanation generation process itself and uses blinded evaluators to assess explanation quality across clarity, usefulness, and contextual adaptation. The results serve as a validation for justifying the design decisions adopted in our approach.

The experiment is based on ten representative smart home scenarios. For each scenario, explanations were generated using four distinct configurations that combine different LLMs and prompting strategies, as summarized in Table 4. Thirty independent evaluators, who were blind to the generation method used, assessed the explanations. The evaluators included university professors, independent professionals, and, in some cases, final-year doctoral students. Their selection was based on their mastery of the subject and their experience in activities directly linked to our research, with the aim of guaranteeing a transparent and reliable evaluation. The evaluations assess three predefined dimensions:

- **Clarity** (1 = not clear, 5 = very clear)
- **Useful** (1 = not useful at all, 5 = very useful)
- **Adapted to the situation** (1 = not adapted at all, 5 = very well adapted).

The average scores obtained for each generation method (condition) are presented in Table 5. The use of structured prompts significantly improves the perceived quality of the generated explanations compared to simple prompts, independently of the underlying language model (GPT-3.5 or Claude 3 Haiku). Moreover, fine-tuning the model with

**Table 6 Evaluation instrument for generated explanations.**

| Negative evaluation | Positive evaluation |
| --- | --- |
| Annoying | Pleasant |
| Complicated | Easy |
| Unusable | Useful |
| Confusing | Clear |
| Boring | Lively |
| Not interesting | Interesting |
| Doubtful | Reliable |
| Out of place | Adapted to the situation |

domain-specific data brings additional gains, particularly in the dimension of contextual adaptation.

Interestingly, although the simple prompts generated by Claude 3 Haiku performed slightly better than those from GPT-3.5, the structured prompting strategy remains the most influential factor in enhancing explanation quality. The best performance was achieved when combining structured prompting with domain-specific fine-tuning. These findings substantiate the design choices made in our approach, confirming that investing effort into both prompt engineering and fine-tuning is justified when aiming to deliver clear, useful, and context-aware explanations in adaptive smart environments.

# EVALUATION OF THE USER EXPERIENCE

To complement the prior validation of our proposal, this section introduces an experiment aimed at illustrating how users perceive the explanations generated by a system implementing our solution. The experiment evaluate the user experience in response to explanations provided by the system. The evaluation was conducted through a survey that collected user feedback for a smart home system. The section describes how the survey was carried out, including its design and execution, and presents the results. This approach allows us to assess the effectiveness of our approach in delivering dynamic explanations that enhance the user experience.

## Questionnarie

We used a questionnaire to collect data about the user experience. The questionnaire is based on the UEQ-S (*Schrepp, Kollmorgen & Thomaschewski, 2023*), which includes eight items. UEQ-S is a short version of the User Experience Questionnaire (UEQ) designed to allow a quick assessment of user experience (UX) (*Schrepp, Thomaschewski & Hinderks, 2017*). UEQ-S is available for scenarios requiring very short completion times. This kind of questionnaires has been used by previous research to assess explanations generated by both LLM and humans, in controlled, non-interactive contexts. For example, *Krause & Stolzenburg (2024)* employed a questionnaire to evaluate ChatGPT-generated explanations in common-sense reasoning tasks, while *Omeiza et al. (2021)* applied a similar methodology based on exposure to sequences of images depicting hypothetical scenarios of autonomous driving, accompanied by system-generated explanations, to assess how

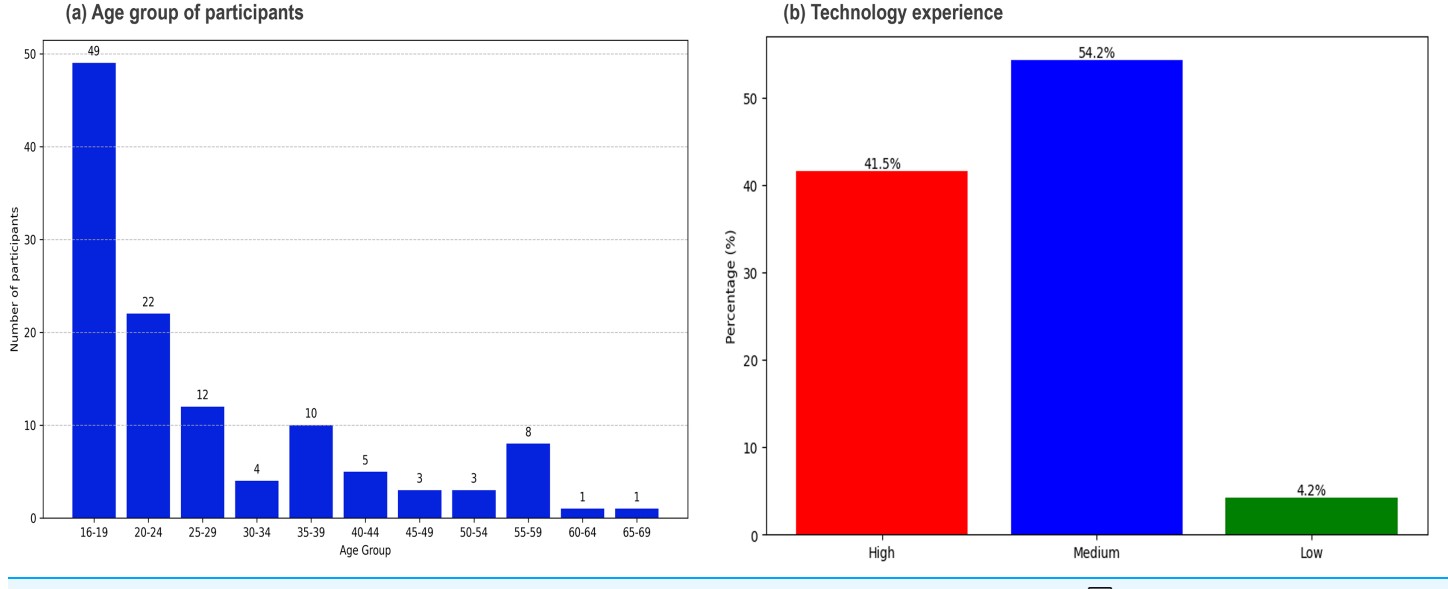

**Figure 6 Distribution of participants and technological experience.**     

different types of explanation influence understanding of events and perceived confidence in vehicle behavior.

Specifically, the questionnaire includes four questions from the pragmatic scales Efficiency, Perspicuity, Dependability, and four questions from the hedonic scales Stimulation and Novelty.

Some items of the UEQ-S were not suitable for our study because we are focused in the adaptability of the explanations to concrete situations than in being at the forefront. Therefore, we replaced them with more precise items. Table 6 shows the issues assessed in our questionnaire. Each issue is defined by the negative evaluation value and positive evaluation value. The eight questions are scored from −3 (most negative evaluation) to +3 (most positive evaluation).

## Participants

We collected responses from 118 people randomly selected by colleagues of the researchers. Data collection on the subjects' background and experience was performed through a demographic questionnaire applied at the first stage of the survey. The questionnaire included two questions to collect personal data (age and technological experience). We aimed to ensure a diverse sample in terms of age and previous experience, to maintain the heterogeneity of the group and minimize potential biases in the results of the experiment. All participants provided informed consent before participating in the study. Based on the responses obtained from the demographic questionnaire, Fig. 6 reveals the following conclusions:

- the participants' ages ranged from 16 to 69 years. There was a high participation of adolescents and young adults, particularly men, in the 16 to 24 age groups, with 35 men and 14 women in the 16 to 19 age group.

**Table 7 Scenarios for the execution of the experiment.**

| N | Scenarios | Generated Explanation |
|---|---|---|
| 1 | It is 10:37 a.m., and you are working in your office, connected to a video conference. Suddenly, the system, through the house speakers, notifies you. | *An order has been generated to replenish the refrigerator, automatically activated when minimum levels are reached.* |
| 2 | You are at home, relaxing in the living room after a long day at work. It is 6:30 p.m., and you have just sat down to watch your favourite TV show. Suddenly, you receive a notification on your mobile phone. | *The irrigation system is suspended because the backyard sprinkler is clogged. Remove debris from the sprinkler and manually restart the system to restore its operation.* |
| 3 | It's a quiet night, and you're getting ready for bed. It's 10:45 p.m., and you've just finished your nightly routine. You're about to turn off the lights and go to bed when you hear a voice prompt through your smart speakers. | *It's time to activate the security system. You need to do it manually before going to bed.* |
| 4 | You are hosting a dinner with friends at your house. It is 7:00 p.m., and guests are starting to arrive. You notice that the house feels a bit colder than usual. Suddenly, you receive a notification on your mobile phone. | *The heating has turned off due to a configuration issue. To reactivate it, turn on the thermostat, manually adjust the settings, and set an appropriate temperature.* |
| 5 | It's a cold winter morning, and you've just woken up. It's 7:00 a.m., and as you leave your bedroom, you feel that the house is colder than usual. You head to the kitchen to prepare breakfast, and suddenly, you hear a voice prompt through your smart speakers. | *Due to the living room window being open, the heating system will not activate. Please close the window for it to function properly.* |

- In the experiment participated 94 men and 24 women. The difference in participation is attributed to factors such as accessibility, interest, or willingness to take part in the experiment.
- Regarding technological experience, the majority of participants fell into the medium category, representing 54.2% of the total. 41.5% of participants had high technological experience, while only 4.2% were classified in the low category.

## Scenarios

The questionnaire immersed participants in five scenarios within a smart home environment. Table 7 presents the scenarios used in the questionnaire, each oriented toward experiencing different functionalities. The services involved in the evaluated scenarios included purchase order confirmation, irrigation control, security systems, management of curtains and blinds, as well as thermostats and climate control. This diversity of services allowed for a more thorough evaluation of explanations across a wide range of conditions.

## Results

Figure 7 shows the results obtained from the collaboration of 118 participants who evaluated the explanations generated from the proposal. The values shown in Fig. 7 represent the average results from the five scenarios evaluated by the participants, indicating whether the generated explanations met the established indicators. This allowed us to observe the following:

- **Annoying *vs.* Pleasant:** 21.0% of users found the explanations somewhat pleasant, and 18.8% rated them as very pleasant. In contrast, 22.0% remained neutral, 25.3% of participants found them very annoying, and 12.9% found them somewhat annoying.

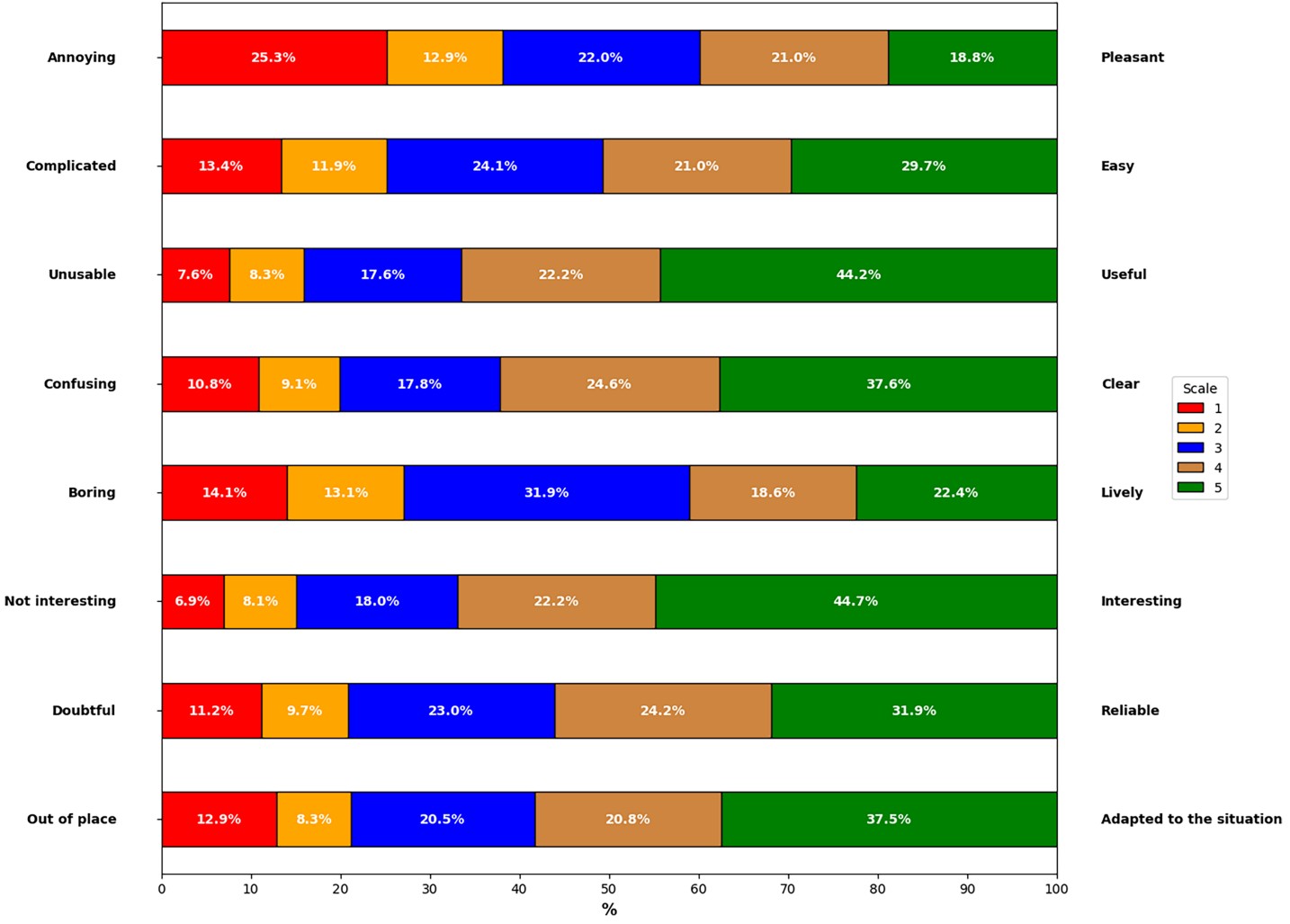

**Figure 7** Results of the questionnaire based on the five scenarios.

- **Complicated vs. Easy:** 29.7% of participants evaluated the explanations as very easy to use, and 21.0% found them somewhat easy. Conversely, 24.1% remained neutral, 13.4% considered them very complicated, and 11.9% found them somewhat complicated.

- **Unusable vs. Useful:** 44.2% of users rated the explanations as very useful, and 22.2% considered it somewhat useful. In contrast, 7.6% found it very unusable, and 8.3% found it somewhat unusable, with 17.6% maintaining a neutral opinion.

- **Confusing vs. Clear:** 37.6% of participants rated the explanations as very clear, and 24.6% found it somewhat clear. However, 10.8% considered it very confusing, and 9.1% found it somewhat confusing, while 17.8% remained neutral.

- **Boring vs. Lively:** 22.4% of users considered the explanations very lively, and 18.6% found it somewhat lively. Meanwhile, 31.9% of participants remained neutral, 14.1% rated it as very boring, and 13.1% found it somewhat boring.

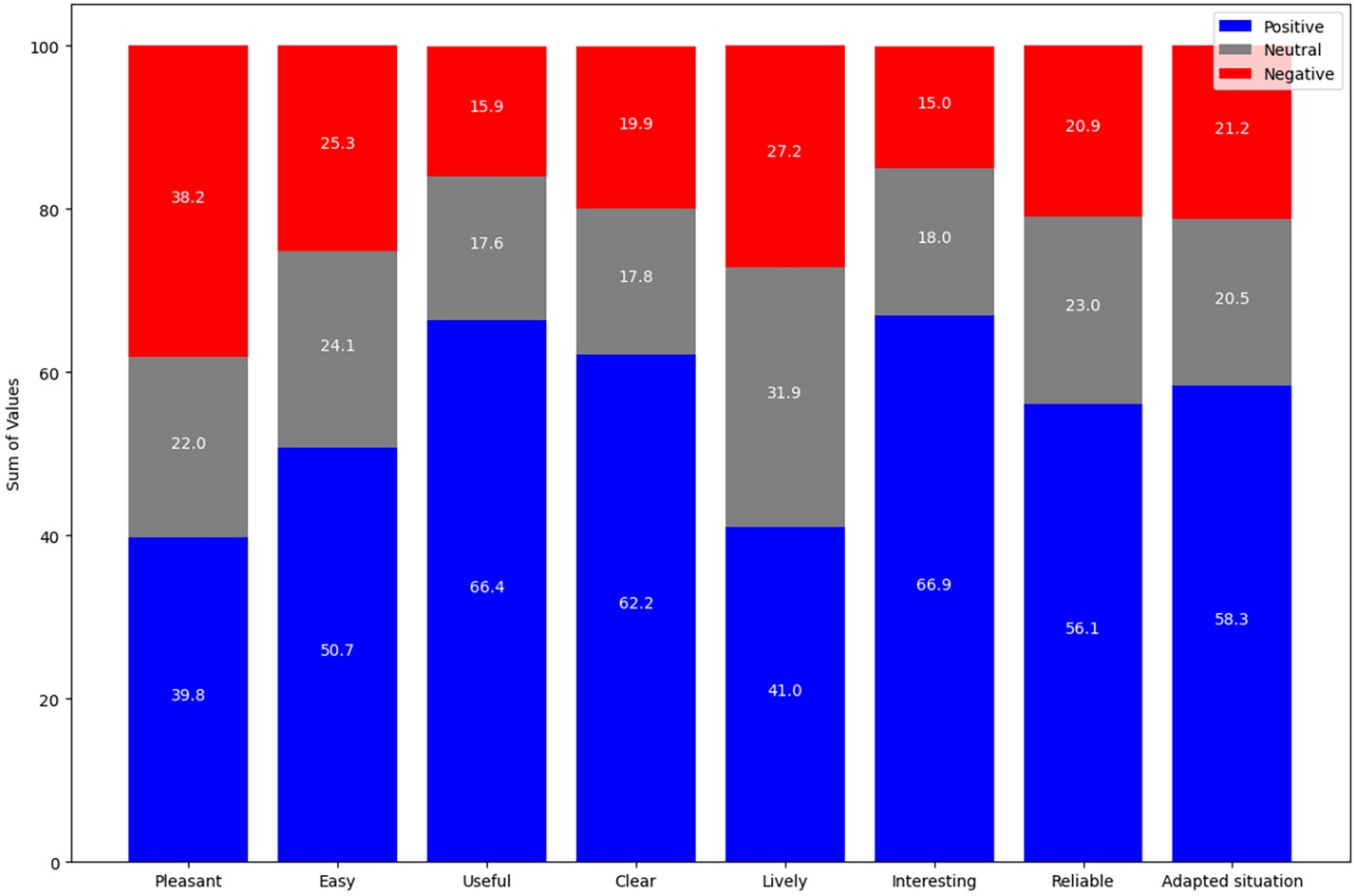

**Figure 8** Evaluation by indicator in negative, neutral and positive scales.

- **Not Interesting *vs.* Interesting:** 44.7% of participants found the explanations very interesting, and 22.2% rated them as somewhat interesting. On the other hand, 18.0% maintained a neutral opinion, 8.1% found them somewhat uninteresting, and 6.9% considered them very uninteresting.

- **Doubtful *vs.* Reliable:** 31.9% of users considered the explanations very reliable, and 24.2% rated it as somewhat reliable. In contrast, 23.0% of participants remained neutral, 11.2% found it very doubtful, and 9.7% found it somewhat doubtful.

- **Out of Place *vs.* Adapted to the Situation:** 37.5% of participants found the explanations very well adapted to the situation, and 20.8% rated them as somewhat adapted. Meanwhile, 20.5% remained neutral, 12.9% considered them very out of place, and 8.3% found them somewhat out of place.

Complementing the above results, Fig. 8 summarizes the evaluation of the explanations in the five smart home scenarios, using a scale from −3 (most negative evaluation) to +3 (most positive evaluation). The "Pleasant" indicator has a response distribution of 39.8% positive, 22.0% neutral, and 38.2% negative, alluding to a moderately pleasant evaluation

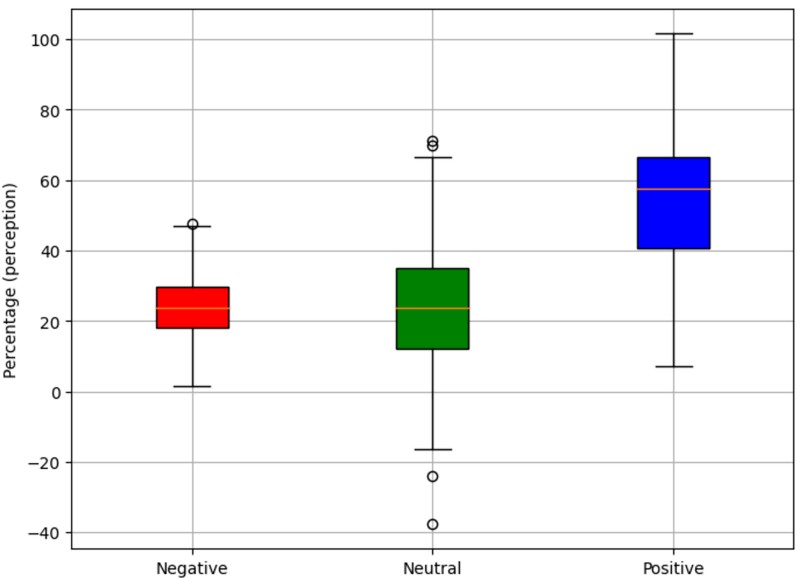

**Figure 9 Perception of the explanations generated.**

with a balance between different responses. The "Easy" indicator shows a higher proportion of positive responses at 50.7%, followed by 24.1% neutral and 25.3% negative responses, indicating an overall ease in understanding the explanations. "Useful" has a high proportion of positive responses at 66.4%, while neutral and negative responses are lower, at 17.6% and 15.9%, respectively, indicating that the explanations are seen as useful. In "Clear", the majority of responses are positive at 62.2%, with 17.8% neutral and 19.9% negative, indicating clarity in the explanations. The "Lively" indicator has a predominance of neutral responses at 31.9%, with 41.0% positive and 27.2% negative, reflecting a less dynamic appreciation of the explanations.

While the indicator "Interesting" stands out with a high positive rating of 66.9%, with 18.0% neutral and 15.0% negative responses, indicating considerable interest from the participants in the explanations. "Reliable" shows a predominance of positive responses at 56.1%, with 23.0% neutral and 20.9% negative responses, meaning the explanations are seen as reliable. The "Adapted Situation" indicator has 58.3% positive responses, 20.5% neutral, and 21.2% negative, demonstrating that the explanations are well-adapted to the user's context. In general, the data reflect a positive response to the explanations generated in all indicators. The relatively small differences between the approval rates for the different indicators could imply that the explanations maintain a consistent level of quality in all aspects evaluated.

In the same direction, Fig. 9 shows a consolidated summary of the results obtained, highlighting a positive perception of the explanations generated by the proposal, with 55.2% acceptance. The 22.9% of the appraisals were neutral. However, a margin for improvement of 21.9% is identified, which represents an opportunity to refine and enrich the current approach.

# LIMITATIONS

The results obtained in the experiment have shown that the explanations produced by our solution are relevant and adequately fit the type of explanation required. However, the present proposal, although innovative and applicable in the field of autonomous systems for smart homes, has certain limitations that must be recognised and addressed in future research and development.

One important limitation of this study is the use of the UEQ-S questionnaire to evaluate explanation quality in static, non-interactive scenarios. Although UEQ-S is a validated and efficient instrument for assessing user experience, it was originally designed for interactive systems and may not fully capture the nuances of explanation utility, timing, or appropriateness when users are not actively engaging with a live system. While this choice allowed us to collect standardized feedback from a diverse sample in an early-stage evaluation, future studies should incorporate more immersive and dynamic methods such as think-aloud protocols, semi-structured interviews, or interactive prototypes that simulate real-time explanation delivery. These approaches would provide deeper insights into user understanding, trust formation, and context-sensitive interpretation of explanations. Also, another limitation in the validation was to validate the timing and manner of delivery of the explanations, given that the evaluation was carried out by means of questionnaires applied in hypothetical scenarios. Some users noted that receiving explanations *via* loudspeaker could be uncomfortable in a real-world setting. Therefore, in subsequent phases, we plan to adjust interaction mechanisms and validate the timing of delivery, as well as incorporate more immersive evaluation methods, as mentioned before.

Another limitation is related to the level of complexity addressed in the evaluation scenarios. In this initial phase of the study, we chose to work with simple cases, characterized by straightforward adaptation structures and bounded conditions, in order to test, in a controlled environment, the technical feasibility of the approach and its ability to generate explanations in the smart home domain. However, the system does not yet address situations involving multiple simultaneous rules, overlapping contextual conditions, or composite explanations integrating several actions. Future studies are expected to address these scenarios where multiple adaptations may occur concurrently, which will be relevant to evaluate the scalability of the system and its performance in contexts of higher complexity. Although the current proposal does not yet implement these mechanisms, the system architecture is modular and allows for the integration of:

- a reasoning layer to identify and consolidate multiple causal factors.
- A conflict-resolution strategy to select the most informative or safest explanation.
- Techniques for generating multi-causal or summary-level explanations when appropriate.

In the same direction, another limitation concerns the feedforward component of the generated messages. The system does not currently perform diagnostic reasoning or error analysis, but rather relies on general patterns learned during fine-tuning to provide action suggestions. However, this reliance limits its ability to generate more complex

explanations, such as recommending configuration changes or resolving system errors. While this often results in helpful guidance, particularly in routine situations, it may lead to generic instructions when detailed knowledge about device capabilities or failure states is unavailable. Future work will explore integrating structured knowledge sources to enhance the reliability and contextual relevance of such explanations.

Furthermore, the need to incorporate more sophisticated feedback mechanisms that adjust the explanations according to the user's response is recognized. In this line, emotional inference by analyzing voice, facial expressions, or gestures could provide valuable information on the understanding or acceptance of the explanations, favoring a dynamic and non-intrusive adaptation of the interaction.

A further important consideration is the inherent risk of LLMs generating hallucinations, *i.e.*, explanations that are plausible but do not correspond to the actual state of the system. Although fine tuning with domain data was applied in our proposal, this phase of the study did not contemplate specific mechanisms to detect or mitigate this phenomenon. In this context, where the reliability of the explanations is critical, the incorporation of supervised fine-tuning techniques that penalize hallucinations during the generation of explanations (*Song et al., 2024*), and recovery augmented generation (RAG) strategies that allow anchoring the explanations in verifiable and system-specific sources are proposed as a future line of research.

Similarly, the use of LLMs managed by third parties poses significant user privacy challenges. In standalone systems, sending data to external services may pose a risk of unwanted exposure of sensitive information. While this phase of the study was conducted using simulated scenarios, it is recognized that future implementations need to adopt more secure strategies, such as local processing, deployment of open-source models at the edge, and application of anonymization techniques. Also, it is suggested to work closely with legal and ethical specialists to ensure compliance with regulations such as GDPR and promote more responsible interactions.

In addition, the integration of a customer effort score (CES) and customer satisfaction score (CSAT) module would facilitate the possibility of measuring the simplicity of processes and the reduction of effort required from users. This implementation would not only minimize complications during interaction with the system but would also provide valuable indicators on the level of overall satisfaction, clarity of explanations, and ease of use. By way of example, these could be some of the questions: "Did you find the explanations provided by the system helpful and easy to understand?", "Were you satisfied with the speed with which the system responded to your needs?" and "What additional functionalities would you like the system to incorporate to enhance your user experience?". Collecting this data could improve the design and refine the explanations and preferences the system presents to users.

Finally, although the proposed solution has been instantiated in the smart home domain, it impacts not only this domain, nor just the autonomous systems domain, but also has potential applications in several other areas, including digital health, industrial automation, energy management, and agriculture. In the digital health domain, for example, it would facilitate personalized explanations to patients about their treatment or

the use of medical devices, improving adherence and understanding of healthcare. A specific example would be the system explaining to patients how to take their medications correctly and what side effects to observe, adjusting the explanations according to the patient's needs. In industrial automation, tailored explanations would optimize operator training and process management, increasing efficiency and reducing errors. For example, the system could provide detailed real-time instructions on how to perform maintenance on specific machinery, explaining each step of the process clearly and concisely and adjusting the explanations according to the operator's experience. In energy management, it would provide customized explanations for companies' efficient use of resources. For example, the system would explain in detail how to optimize the use of machinery and lighting systems in a production plant, providing clear instructions based on the company's energy usage patterns and current operational demands. In agriculture, it would provide timely information on crop management, irrigation optimization, and pest control, resulting in increased productivity and more sustainable agricultural resource management. The system could explain to farmers how and when to apply fertilizers based on real-time data on soil and crop conditions. We interpret these results as a first validation of the path we have opened in this work, in which we use language models to generate explanations tailored to users' preferences and needs.

## CONCLUSIONS

This article has tackled the challenge of generating explanations at run-time within autonomous systems, driven by the necessity of user comprehension and trust for effective human-system collaboration. Our proposed solution focuses on generating the content of these explanations dynamically. We leverage the power of large language models (LLMs), which are fine-tuned with domain-specific data and prompts, enabling the generation of dynamic explanations tailored to user context. We demonstrate the solution in the domain of smart homes by using ChatGPT, a LLM to create these adapted explanations. Through this approach, we ensure that explanations are not only comprehensible and trustworthy but also relevant, thereby promoting a seamless interaction between humans and systems. To validate the generated explanations, we conducted an experiment designed to assess the user experience with the system-generated explanations within a simulated smart home environment.

Our approach goes beyond traditional static methods by integrating contextual awareness and the user profile into the explanation generation process, ensuring that explanations are continuously updated and aligned with both user expectations and system behavior. In doing so, we address a crucial gap in current research—delivering dynamic, user-specific explanations that enhance trust, transparency, and collaboration in autonomous systems.

Although the evaluation was conducted using predefined scenarios, it is important to note that the explanation generation technique—comprising the construction of prompts, integration of contextual variables (*e.g.*, user profile, environmental data, interaction history), and the generation of natural language responses by a fine-tuned LLM—has been fully implemented and is operational within a simulated smart home environment. This

setup enables real-time explanation generation in response to simulated adaptation events, allowing the system to function end-to-end for the purposes of testing and validation. While a full deployment in a live smart home system remains future work, the core components required for explanation generation are functional and have been validated through the scenario-based evaluation.

Moving forward, our work sets the stage for the development of more transparent and trustworthy autonomous systems. Future research avenues include:

- exploring the synergies between our solution and other human-centered design principles for autonomous systems.
- Assessing the generalizability of our approach across diverse application domains.
- Investigating user acceptance and preferences regarding explanations generated by LLMs.

Addressing these future directions will further refine the capabilities of autonomous systems, enabling them to effectively collaborate with humans and address complex challenges in a myriad of contexts.

### Funding
This work was developed with the support of the Generalitat Valenciana with TENTACLE (CIAICO/2023/089), and the Spanish Ministry of Science and Innovation under the Project PRODIGIOUS PID2023-146224OB-I00. The funders had no role in study design, data collection and analysis, decision to publish, or preparation of the manuscript.

### Grant Disclosures
The following grant information was disclosed by the authors:
Generalitat Valenciana with TENTACLE (CIAICO/2023/089).
Spanish Ministry of Science and Innovation: PRODIGIOUS PID2023-146224OB-I00.

### Competing Interests
The authors declare that they have no competing interests.

### Author Contributions
- Oscar Peña-Cáceres conceived and designed the experiments, performed the experiments, analyzed the data, performed the computation work, prepared figures and/or tables, authored or reviewed drafts of the article, and approved the final draft.
- Antoni Mestre analyzed the data, performed the computation work, prepared figures and/or tables, authored or reviewed drafts of the article, and approved the final draft.
- Manoli Albert conceived and designed the experiments, performed the experiments, analyzed the data, prepared figures and/or tables, authored or reviewed drafts of the article, and approved the final draft.

- Vicente Pelechano conceived and designed the experiments, analyzed the data, authored or reviewed drafts of the article, and approved the final draft.
- Miriam Gil conceived and designed the experiments, performed the experiments, analyzed the data, performed the computation work, prepared figures and/or tables, authored or reviewed drafts of the article, and approved the final draft.

## Data Availability

The data are available at Zenodo: Peña-Cáceres, O., Mestre, A., Albert, M., Pelechano, V., & Gil, M. (2024). Automatic Generation of Explanations in Autonomous Systems: Enhancing Human Interaction in Smart Home Environments [Data set]. Zenodo. https://doi.org/10.5281/zenodo.14191897.

The data, questionnaire answers and codebook are available in the Supplemental Files.

## Supplemental Information

Supplemental information for this article can be found online at http://dx.doi.org/10.7717/peerj-cs.3041#supplemental-information.

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
