# Peer review of "Automatic generation of explanations in autonomous systems: enhancing human interaction in smart home environments"

_PeerJ Computer Science, doi:10.7717/peerj-cs.3041_

## Round 0.1 · original submission · Major Revisions

The manuscript is well-written and addresses an interesting and timely topic. However, substantial revisions are necessary to address the reviewers’ concerns adequately. A major revision is therefore recommended.

Reviewer 1 ·

Basic reporting

The manuscript is well-written, with professional and technically accurate English throughout. Its structure adheres to the journal’s standards, and the figures and tables are relevant and clear. The introduction provides sufficient context, and the literature is well-referenced. Overall, the manuscript meets the journal’s criteria for clarity, structure, and technical quality.

The related works section is also strong, providing a comprehensive overview of the relevant literature in explanation generation, context-aware explanation, and usage of LLM in explanation. However, to further enrich this section, I suggest including references to related works in software architecture and software engineering as they pertain to integrating explanation modules in autonomous systems and smart spaces. Given that explanation systems often operate as part of larger frameworks in such domains, addressing these aspects would provide a broader and more practical perspective on the challenges and solutions for integration.

While the manuscript is commendable in many respects, one notable issue is its ability to stand as a fully self-contained unit of publication. The study is positioned as an extension of the author’s prior work, yet it relies heavily on that earlier research without sufficiently summarizing its foundational elements. This compromises the accessibility and clarity of the present manuscript, as readers are required to consult previous works to understand the current study fully.

To address this issue, I recommend adding a brief but comprehensive summary of the key mechanisms, processes, and findings from the prior works. This could be incorporated into the introduction or methodology section. The summary should cover the foundational aspects that are essential for understanding the current study, including the explanatory model, its motivation, and the contexts previously explored.

Experimental design

The research question targeted in this work—real-time explanations for autonomous systems and smart spaces, with the use of large language models (LLMs) for generating such explanations—addresses an important and timely topic. As autonomous systems and smart spaces continue to grow more advanced and prevalent, the need for effective, real-time explanations becomes increasingly critical. The study’s focus on leveraging LLMs for this purpose represents a novel and valuable contribution to the field.

The scientific approach employed in this study is methodologically sound and demonstrates a commendable effort to address the challenges associated with real-time explanation generation and LLM application in such a domain. While the investigation is valid and provides meaningful insights, the approach could benefit from further refinement and maturity to fully realize its potential. Additional details on these areas for improvement are outlined in the "Additional Comments" section.

Validity of the findings

"All underlying data have been provided; they are robust, statistically sound, & controlled":

The manuscript outlines a rigorous approach for collecting a quite large collection of 228 scenarios in smart spaces where explanations are deemed necessary. This collection represents a valuable contribution to the field, addressing a critical gap in real-world datasets and benchmarks for explanation systems. While I have critiques about the collection process and the systematic nature of the approach, I still believe that this work provides a valuable resource for the research community as it could serve as a valuable resource for developing, testing, and comparing explanation systems in autonomous environments.

I strongly encourage the authors to make these scenarios publicly available.

Additional comments

1. Integration into Host Systems:
While the paper primarily focuses on explanation generation, it is important to consider that such systems are not standalone; they are designed to operate within larger ecosystems, such as smart spaces. To evaluate the feasibility of the proposed approach, the manuscript should clarify how the system can be practically integrated into the host environment. This includes providing insights into the overall software architecture, data flow, and data management necessary for deployment.

The explanation generation process relies on a proposed prompting mechanism that depends on critical information, such as the explanation’s urgency, human context (e.g., activities like working or relaxing), and the initial identification of the need for an explanation. A clearer discussion of how this information is gathered and operationalized in real-world scenarios would strengthen the paper’s relevance and demonstrate the practicality of the approach.
* * *
2. Complexity of Autonomous Systems :
The proposed solution appears to underestimate the inherent complexity of autonomous systems. In all the scenarios presented, there is a single adaptation action to be explained, which is directly linked to one adaptation rule. However, real-world cases are often far more intricate, involving conflicting rules, overlapping contexts, and multiple interdependent conditions.

For instance, what happens when there are multiple preconditions that could result in a particular action? How does the system prioritize or handle conflicting explanations? Similarly, how does the system manage situations where an explanation needs to address multiple actions or rules simultaneously? Addressing these complexities and providing a discussion on how the system scales to handle such scenarios would significantly enhance the robustness and applicability of the proposed approach.
* * *
3. Issues with the Feedforward Component of Explanations:

The “feedforward” part of the explanation raises significant concerns in the manuscript. While the “feedback” part is straightforward—typically stating the precondition and action of the event—the construction of the feedforward explanation is unclear and poorly detailed. The manuscript does not explain whether the inference about what to do next is entirely generated by the LLM. If that is the case, is it truly reliable to rely solely on a language model to deduce the next steps without providing structured data about the system’s logic, device behavior, or operational context?

In the provided examples, the feedforward explanations often seem trivial and, essentially, the negation of the precondition or action. For instance:
• “The system turned off the heating because it detected a problem with the thermostat. To turn it back on, you need to manually reset the thermostat.”
This reduces to: “Heating is off because of the thermostat --> Turn on the thermostat!”
• “The main sprinkler is off due to pressure above 50 PSI. Manually adjust the pressure valve until it drops below 50 PSI and restart the irrigation system to resume operation.”
This translates to: “Sprinkler is off because pressure > 50 PSI --> Lower the pressure below 50 PSI.”

These examples highlight that the next actions are often trivial reversals of the conditions, raising the question of whether explanations are even needed for such cases. If the next step is so obvious, does the explanation system add meaningful value?

Furthermore, there are cases where the feedforward explanation suggests modifying a device’s settings, such as:
• “The alarm was triggered by movement from the guest bedroom window sensor. No suspicious activity was detected. Adjust the sensor’s sensitivity to avoid false alarms.”

In this case, it is unclear how the system “knows” that the sensor has adjustable sensitivity levels. Does the explanation system rely entirely on the LLM to generate this information without validating whether the device in this specific smart space actually supports such a feature? If so, this raises concerns about hallucination by the LLM. Alternatively, does the explanation generation system have access to metadata about the devices and their functionalities? If the latter is true, it circles back to point 1, as it is not clear how such metadata is gathered, structured, and provided to the explanation system systematically.

Another critical concern on "feedforward" is how the system handles cases of internal failures or errors. For instance, if a behavior results from a system malfunction, does the explanation system engage in “debugging” to trace the issue to its root cause and provide precise feedforward instructions for error resolution? If so, how does it gather the necessary information, and wherein the “prompt structure” is this data incorporated? Alternatively, does the system merely provide generic error-handling advice, such as: “The sprinklers are off due to error number X. To resolve it, address error X.” ?

Without a systematic method for monitoring, diagnosing, and updating information about devices and their operational states, the feedforward component lacks reliability and scalability. Addressing these issues would require a detailed discussion of how the system gathers and processes the information needed to generate actionable and contextually valid feedforward explanations.
* * *
4. Issues with the Prompt Design and Explanation Type:

A major concern lies in the design of the prompt itself. According to the manuscript, the explanation is supposed to be either “feedforward” or “feedback,” as indicated by the inclusion of the Type field in the prompt structure. However, examining the provided scenarios reveals that almost all explanations inherently include a “feedback” component, even when the explanation is classified as “feedforward.” This appears to be a result of the structure of the prompt itself, which enforces the inclusion of a feedback element.

For instance, the explanations consistently start with phrases like:
”[For {Action} triggered by {Event}]…”

This phrase inherently describes the “what” part of the explanation—essentially a statement of what the system just did and why it occurred. By definition, this constitutes the feedback component of the explanation. Thus, while the manuscript describes the approach as distinguishing between “feedback” and “feedforward” explanations, the prompt structure ensures that feedback is always included in every explanation, leading to a lack of alignment between the stated approach and its implementation.

Furthermore, the provision of the content of the explanation (e.g., “What: What has the system done?”, “Why: Why did the system do X?”, “How to: How can I get the system to do Z?”) is inherently tied to the type of explanation. Specifically:
• Feedback explanations typically answer “What” and “Why.”
• Feedforward explanations, by definition, would not address “What” (since that pertains to past actions) but focus on “How to.”

This leads to an inconsistency in the prompt structure: a feedforward explanation cannot logically answer the “What” question, yet the current prompt framework suggests it should. This inconsistency is evident in the generated explanations, where the “What” part is essentially feedback, even when the explanation is supposed to be feedforward. The feedforward component then either becomes a trivial negation of the event or a random, unvalidated suggestion from the LLM.

To address these inconsistencies, the authors need to refine the prompt structure to better align with the intended distinction between feedback and feedforward explanations. Additionally, there should be clarity on how the type and content of the explanation are systematically defined and generated to avoid these contradictions.
* * *
5. Validation of the Proposed Solution:

The validation of the proposed solution is a significant weakness in the manuscript. The current evaluation approach is not only irrelevant in some aspects but also disproportionately narrow compared to the scope of what has been proposed. The authors miss the crucial “evaluation” point: the question is not simply, “Here is an explanation the system generates—do you like it?” Instead, the focus should be on assessing whether the characteristics of the explanation, such as being context-aware or using an LLM trained on domain-specific data, have a tangible impact on producing better explanations.

What should have been evaluated is a comparison between the explanations generated by the proposed system and those generated using alternative methods. For example:
• What if explanations were generated using a much simpler prompt (e.g., containing only the description of the adaptation action)? Would the explanations produced by the current system show meaningful improvement?
• What if a generic LLM were used instead of one trained on domain-specific data? Is the time and resource investment in creating domain-specific datasets truly justified?

Such questions are critical for understanding the validity and value of the proposed solution. They can only be answered through comparative studies that benchmark the proposed system against simpler or state-of-the-art explanation generation approaches.

I strongly recommend that the authors conduct a study where explanations are generated using different approaches (e.g., simple prompts, generic LLMs, and domain-specific models). The results should then be compared to assess whether the explanations produced by their system genuinely perform better. Without this comparative evaluation, the manuscript fails to substantiate the claims about the advantages of the proposed system, leaving the overall contribution questionable.
* * *
6. Gender as a Factor in Explanation Generation

The inclusion of “gender” as part of the user profile to influence the generation of explanations strikes me as highly questionable. It is unclear how a user’s gender would shape the type of information or explanation they receive. In fact, this design choice raises concerns about potential violations of gender equality principles, as it suggests engineering a system to tailor explanations based on gender. I strongly recommend that the authors consult with specialists in ethics, gender studies, or human-computer interaction to critically assess whether such a feature is appropriate or necessary.
* * *
7. Avoiding Overstatements in Contributions:
The manuscript tends to overstate its contributions in a way that can be misleading. For example, on page 5, three bullet points outline the capabilities of the proposed solution, but the subsequent paragraph clarifies that the manuscript only focuses on Step 2. This seems unnecessary and overstated. I suggest consolidating the bullet points to highlight Step 2 directly, with the accompanying text noting that the other steps are handled in preceding or subsequent work.

Similarly, in Figure 1 and the accompanying text on explanation content, the authors list multiple possibilities such as “why,” “what,” “why not,” and “how to.” However, the proposed solution only addresses “why” and “what,” which are the most straightforward among these. More complex types of explanations like “why not” and “how to” are not supported by the current prompt engineering. This limitation should be clearly acknowledged in the manuscript to avoid misrepresenting the scope of the contribution.

Reviewer 2 ·

Basic reporting

The authors propose a solution for the automatic generation of explanations to improve computer-human interactions in smart home environments. The paper describes a conceptual model for specifying explanations and proposes an approach based on large language models (LLMs) tuned with domain-specific data. The application of this approach to a smart home system is illustrated on a specifically created dataset and the refinement of ChatGPT to produce contextually relevant explanations. Finally, a validation through a user experience survey is presented, the results of which show a generally positive perception of the generated explanations.

The paper is clear and well structured and it has the writing soundness necessary for a scientific publication. Related works are properly described and the paper follows a clear path from the hypothesis (explanations to improve interaction) to the conclusion (results from a survey).

Experimental design

Concerning the experimental design I have the following three concerns that the authors might consider to tackle:
1. One of the critical challenges in using LLMs for explanation generation is the phenomenon of hallucinations. Although the authors fine-tune GPT-3.5 with domain-specific data, the paper does not address whether or how this process mitigates hallucinations. A discussion of this issue is essential, especially in the context of autonomous systems where trust and reliability are crucial. At the very least, the potential for hallucinations should be acknowledged as a limitation, with suggestions for future work on mitigation strategies.
2. The paper builds its approach on GPT-3.5. Given the rapid evolution of open-source and commercial LLMs (e.g., Mistral, LLaMA, Claude, Gemini), it would strengthen the work to explore the rationale for this choice even more in deep that what the author wrote, for example considering factors such as performance, accessibility, fine-tuning capabilities, or integration constraints.Of course, I am not asking to add more tests, but just to strengthen the explanation.
3. The proposed approach appears to rely on deploying or accessing LLMs such as GPT-3.5, which are typically hosted and operated by third-party providers (e.g., OpenAI). This raises significant concerns regarding the privacy and security of user data in smart home environments. The paper should discuss the implications of transmitting user-generated content to external services, particularly in light of applicable privacy regulations (e.g., GDPR), and include this as a limitation of the current approach.

Validity of the findings

The validity of the proposed approach is based on a survey of 118 participants who evaluated explanations in a smart home environment . Participants were presented with five different scenarios and used a modified UEQ-S questionnaire to provide feedback . The survey aimed to assess the user experience with the system's generated explanations. Being this based on relevant literature, I think it is adequate and I have no additional comments on the validity of the findings.

Reviewer 3 ·

Basic reporting

The paper titled "Automatic Generation of Explanations in Autonomous Systems: Enhancing Human Interaction in Smart Home Environments" is about automatic real-time explanation generation with the intention of clarifying dynamic system behavior to users. The topic has a very practical approach and relevance for everyday use. The focus on the interaction between users and autonomous systems also aligns with the growing need for transparency and trust in AI-driven systems. The use of LLMs (e.g., GPT) for explanation generation is a novel direction and contributes to ongoing discussions about explainability in applied contexts. However, the current submission presents several structural and methodological limitations that hinder the clarity and impact of its contributions. Below, I detail key areas for improvement.

I recommend a rewrite of the abstract. The current abstract starts very general, and after reading the first half, the domain and specifics of the paper are unclear. In particular, it starts very broadly while the contribution of the paper is focused.

The paper begins with a broad framing of explainability, yet its contributions are focused on a specific scenario: user-facing explanation in smart home contexts. I strongly recommend reworking the abstract and introduction to reflect this scope more directly. Reading the paper, I first expected a different kind of explanation. Given that the explanation has broad discussions, I recommend narrowing it down early.
This reframing would not only clarify the problem being addressed but also help distinguish the work from broader explainability or ML explanation literature. For relevant examples, see:
https://dl.acm.org/doi/10.5555/3721488.3721795
https://ieeexplore.ieee.org/abstract/document/9196299
https://dl.acm.org/doi/full/10.1145/3666005

The related work helps narrow down the topic very much; I recommend focusing entirely on smart homes and discussing different types of explanations and provide a reader an overview of relevant work more comprehensively.
for examples: https://dl.acm.org/doi/full/10.1145/3561533
http://link.springer.com/chapter/10.1007/978-3-031-63803-9_23
https://doi.org/10.1145/3517224

Experimental design

The paper claims to enable "automatic real-time explanation generation," but the current system is more exploratory and conceptual than operational. The evaluation is scenario-based and lacks real-time deployment. It would be helpful to clarify which components are implemented versus conceptual and explicitly specify the work’s contribution boundaries. For instance: is the main contribution the conceptual model, the adaptation of GPT, or the experimental design?

The proposed conceptual model (Table 1) raises important questions:
* How was it developed? Was it derived from empirical studies, literature synthesis, or expert interviews?
* What methodology underpins the selection of elements, and how are they justified?
* Are any parts of the model novel in the context of smart home systems?
* Why are the selected elements (e.g., Table 1) not other elements?
Without a clear grounding, the model can be confusing. If it is to serve as a foundation for future system design, its construction must be more rigorously motivated and validated.

Validity of the findings

The use of the UEQ to evaluate explanation quality in static, hypothetical scenarios is not sufficiently justified. UEQ is typically applied to interactive systems, and its use here may not capture the nuances of explanation utility, clarity, or appropriateness.

The lack of real user interaction with a working system significantly limits the generalizability of the findings.

A more appropriate evaluation approach could involve think-aloud protocols, semi-structured interviews, or even mockup interactions that simulate explanation requests in real-time.

Additional comments

Taken together, I recommend reframing the paper's scope, clarifying the conceptual model's origin (literature or evaluation if it is original, or e.g., design workshop, expert elicitation), improving hte evaluation design, and expanding the related work section. A discussion of limitations is also encouraged. In addition, I could not find information related to the IRB of the study.

---

## Round 0.2 · accepted · Accept

The authors have addressed all of the reviewers' comments. The manuscript is ready for publication.

Reviewer 1 ·

Basic reporting

The manuscript has been substantially improved, and the raised issues have been adequately addressed.

Experimental design

The manuscript has been substantially improved, and the raised issues have been adequately addressed.

Validity of the findings

The manuscript has been substantially improved, and the raised issues have been adequately addressed.

Reviewer 2 ·

Basic reporting

The authors tackled the issues that I remarked in my previous review. Everything is ok now.

Experimental design

No comment.

Validity of the findings

No comment.